# Geomorphological evolution of landslides near an active normal fault in Northern Taiwan, as revealed by LiDAR and unmanned aircraft system data

Kuo-Jen Chang[1], Yu-Chang Chan[2], Rou-Fei Chen[3], Yu-Chung Hsieh[4]

[1]Department of Civil Engineering, National Taipei University of Technology, Taipei, 10654, Taiwan, R.O.C.
[2]Institute of Earth Sciences, Academia Sinica, Taipei, Taiwan, R.O.C.
[3]Departement of Geology, Chinese Culture University, Taipei, 11114, Taiwan, R.O.C.
[4]Central Geological Survey, MOEA, Taipei, Taiwan, R.O.C.

*Correspondence to*: Kuo-Jen Chang (epidote@ntut.edu.tw)

**Abstract.** Several remote sensing techniques, namely traditional aerial photographs, an unmanned aircraft system (UAS), and airborne LiDAR, were used in this study to decipher the morphological features of obscure landslides in volcanic regions and how the observed features may be used for understanding landslide occurrence and potential hazard. A morphological reconstruction method was proposed to assess landslide morphology based on the dome-shaped topography of the volcanic edifice and the nature of its morphological evolution. Two large-scale landslides in the Tatun volcano group in Northern Taiwan were targeted to more accurately characterize the landslide morphology through airborne LiDAR and UAS-derived digital terrain models and images. With the proposed reconstruction method, the depleted volume of the two landslides was estimated to be at least $820 \pm 20 \times 10^6 \, \mathrm{m}^3$. Normal faulting in the region likely played a role in triggering the two landslides, because there are extensive geological and historical records of an active normal fault in this region. The subsequent geomorphological evolution of the two landslides is thus inferred to account for the observed morphological and tectonic features that are indicative of resulting in large and life-threatening landslides, as characterized using the recent remote sensing techniques.

Keywords: Unmanned aircraft system, LiDAR, Digital terrain models, Landslide characterization, Morphological reconstruction, Normal faulting, Geomorphological evolution

## 1 Introduction

Landslides pose long-lasting threats to humans and their property and are detrimental to the environment in general. Therefore, many efforts have been made to assess landslide hazards and propose mitigation methods based on key

landslide characteristics, including range and extent, volume, triggering mechanism, recurrence, and subsequent evolution. The overview of landslides provided by morphological analysis has been fundamental in improving the understanding of landslides. Several useful morphological characteristics have been proposed for identifying potential landslide sites (Varnes, 1978; IAEG, 1990; Guzzetti et al., 2012; Santangelo et al., 2015a and 2015b; Bucci et al., 2016); these can be readily applied for landslide recognition and identification. Clayton, Simons, and Matthews (1982) suggested a useful procedure for improving site investigations and methods. Such methods may facilitate the identification of landslide sites and subsequent analysis of landslides. However, before regional-scale landslide recognition and identification, informative morphological analysis must be performed, the results of which are dependent on the quality and precision of the data set. Morphological analysis involving observation and identification methods, such as contour topographic maps, aerial photography, and remote sensing images, is considered an essential task in the early stages of landslide investigations (IAEG, 1990; Santangelo et al., 2015a and 2015b).

The morphological features of relatively old landslides are often perturbed by surficial processes, such as weathering, erosion, and vegetation colonization. The limited precision of aerial photographs and satellite images in many cases prevents detailed analysis of old landslides. Landslide recurrence is a major concern in engineering projects because it reveals the evolution of a landslide and its potential activity. However, landslide recurrence at a specific site cannot be easily and precisely determined, primarily because of the lack of georeferenced historical records of recent landslides. Therefore, it is important to study landslides that date from an earlier period in the evolutionary history of an area.

Volume estimation is crucial for characterizing and understanding landslides. The cut-and-fill volumes of a landslide can be estimated using two digital terrain models (DTMs): one before the landslide and one thereafter (Chang et al., 2005; Chen et al., 2006; Chan et al., 2012, Hsieh et al., 2016). The information derived using the DTMs can facilitate the identification and estimation of landslide activity and evolution. However, estimating the volumes of an old landslide site is relatively challenging. A method for determining landslide volume is to reconstruct paleogeomorphology before a landslide, which requires regional detailed topographic data and reasonable methods of geomorphological reconstruction.

Large-scale landslides are common and frequently recurrent in volcanic regions. The study area, Jinshan, is part of a volcano group in the Taipei metropolitan area in Northern Taiwan (Fig. 1). Two large-scale landslides have already been suggested by C. T. Lee of National Central University (personal communication) from LANDSAT images and 40m grid digital terrain model of the region but without formal documentation in the literature. Because of a lack of relevant information, the details of these landslides are yet to be analyzed. In this study, due to recently developed high resolution and high precision airborne light detection and ranging (LiDAR) and unmanned aircraft system (UAS) datasets, we

proposed feasible methods for evaluating landslides, particularly old landslides, by performing paleotopographic reconstruction based on several concepts and the hypothetical shape of the natural edifice of a stratovolcano. The reconstructed instances of paleotopography were evaluated using different tests and used in the subsequent landslide analysis, including analysis of the landslide volumes and morphological evolution. Moreover, the interactions of tectonics and surface processes, including normal faulting, riverbed incision, and river drainage formation, are discussed in the context of the old landslides to provide perspectives on the newly acquired detailed topography.

## 2 Geological background

Taiwan is situated on an active orogenic belt formed by oblique convergence between the Philippine Sea and Eurasian plates (Ho, 1986; Malavieille et al., 2002). Despite the ongoing collision in Central and Southern Taiwan, a postcollision extension regime has developed since the Plio-Pleistocene in the northern part of this orogeny (Teng, 1996) and has generated volcanic activity both onshore and offshore in Northern Taiwan (Wang et al., 1999). The study area is composed of late Paleogene to Neogene sedimentary rocks (Suppe, 1984; Ho, 1986). The belt is defined by a series of imbricated, trending NE–SW, west-vergent folds and faults (Fig. 1). The strata are composed of late Oligocene to Pleistocene sedimentary rocks and are overlain by the intruded Tatun volcano group.

The study area has two major faults: the Kanchiao fault (KCF) and Jinshan fault (JSF; Fig. 1). The stratigraphic separation of the KCF through thrusting is estimated to exceed 2000 m. However, recent marine seismic survey data clearly indicate that the KCF is now a normal fault. The NE–SW trending JSF, which is situated in the northeast of the Tatun volcano group around Jinshan, has an estimated throw of 3000 m by thrusting. The late Oligocene Wuchihshan Formation is juxtaposed against the late Miocene Nanchuang Formation. It dips over 50° toward the southeast with a trend of N60E°.

These NE–SW-trending thrust faults and folds were identified on land (Huang, 1988), and their offshore extents have been confirmed (Hsiao, 1997; Hsiao et al., 1998). Before a major orogenic movement, the tectonic setting in this area of Taiwan was dominated by normal faults and half grabens (Hsiao, 1997; Hsiao et al., 1998). Faults in this region were reactivated by extensional rifting tectonics as compressional synorogenic fold-and-thrust tectonism during the late Mio-Pliocene period. Postorogenic extension subsequently resulted in the kinematics of these faults reverting to their earlier normal faulting regime. Currently, the offshore structures in northeastern Taiwan are in an extensional regime once again. The JSF has thus reactivated and reverted to being a normal fault, now known as the Shanchiao fault (SCF).

The extensional postorogenic tectonism resulted in extensive igneous activity that began at approximately 2.8-2.5 Ma (Wang, 1989) and continued throughout the quaternary period (Wang, 1989; Tsao, 1994). These volcanic rocks

overlie sedimentary rocks that range from being late Oligocene to early Pleistocene. To a certain extent, the sequence of eruption history can be interpreted by the geomorphological features, petrological characteristics, and geochemical signatures of the volcanic rocks (Wang, 1989; Song, 2000; Chen, 2003). In the Tatun volcano group and its immediate vicinity, most lava flows were dated using K-Ar and fission-track dating methods, which provided ages ranging from 0.8 to 0.2 Ma (Tsao, 1994; Song, 2000).

The Taipei Basin is situated on the other side of the Tatun Volcano and is composed of an asymmetric wedge-shaped half-graben, adjacent to the Linkou Tableland. Both the Linkou Tableland and the basement of the Taipei Basin share the same age of 0.4 Ma (Wei et al., 1998; Teng et al., 2001). The elevation difference between the Taipei Basin and Linkou Tableland is as much as 250 m, meanwhile the depocenter of Taipei Basin is at least 700 m in depth near the east margin of the Linkou Tableland that is seperated by the Shanchiao normal fault (Teng, et al., 2001; Chen et al., 2007; Huang et al., 2007). This phenomenon reveals that the Shanchiao normal faulting resulted in at least one thousand meters of throw of fault separation and contributed to the formation of Taipei Basin.

Recent studies reveal high tectonic activities of the Shanchiao normal fault, including geomorphic and geologic analysis by borehole investigation and record (Chen et al., 2007; Huang et al., 2007; Song et al., 2007), by the geodetic study (Yu et al., 1999), by the micromorphological study (Chen et al., 2006) and by geomorphological study (Chang et al., 2010). The results show that the normal faulting slip rate is estimated between 8.2-1.8 mm/yr in subsidence at different sites and in time intervals. (Huang et al., 2007; Chen et al., 2010). Comparing with the Shanchiao fault, however, the geometric property and tectonic activity of the Jinshan fault have not been well resolved. But the Jinshan fault certainly plays an important role for morphologic evolution within the study area.

## 3 Methods of data acquisition

Geomorphologic analysis is one of the most fundamental tasks for the landslide study, however it depends on the resolution and the quality of dataset used (Santangelo et al., 2015b). Classic techniques, such as aerial photography and stereoscope analysis, were used to extract detailed features of the target area (Santangelo et al., 2015b). Nevertheless, in Taiwan, heavy precipitation induced by the annual northeast monsoon modifies easily the landslide topography. On the other hand, the study region is situated within a national park and preserves dense forest very well. Both effects conceal detailed topography and nearly impossible to study directly from aerial photographs and/or satellite images. Traditional aerial photographs, satellite images, and 40-m grid digital elevation model (DEM), these analyses revealed common morphological features, including circular crowns and depletion zones bounded by topographic scarps on the upslope side within the study area. Two domains of different vegetation densities near the upper part of the sliding areas were

observed to be affected by human activities. Two large circular landslides are marked as the northeastern Xishi landslide (XSL) and southwestern Chinshui landslide (CSL) denoted on Fig. 1. In the coarse DEM, the major geomorphic features of these two landslides are limited, with only two fractured circular crowns being visible. In consequence, much higher resolution dataset is required to decipher more detailed landslide geomorphic components and for landslide investigation.

To acquire high-resolution topographic information, a homemade unmanned aircraft system (UAS) was integrated and used in this study. A UAS, commonly known as a drone, is an aircraft that flies autonomously and has many applications because of its low cost, convenience, and high resolution (Huang and Chang, 2014; Chen et al., 2015; Galarreta et al., 2015; Giordan et al., 2015; Tokarczyk, 2015; Bühler et al., 2016; Deffontaines et al., 2016). The UAS used in this study is a modified version of the Skywalker X8 delta-wing aircraft. It is reinforced by carbon fiber rods and

Kevlar fiber sheets. The UAS was launched by hand; it flew, captured photos, and then glided back to the ground. The whole process was conducted using a preprogrammed flight mission organized by a ground control system and remote controlled by the ground control station. The autopilot system was constructed and modified from the open-source APM (Ardupilot Mega 2.6 autopilot) firmware and open-source software Mission Planner, as transmitted through ground-air XBee telemetry.

To generate a high-resolution digital surface model (DSM), orthorectified mosaic images, and a true three-dimensional (3D) model, 1231 photos were captured by a Sony ILCE-QX1 camera mounted on the UAS. One flight mission of a total distance of 80 km, endurance of 90 minutes, and 250 m of mean height above ground level was organized and implemented (Fig. 2). A total coverage area of 14.7 km$^2$ with a 6.6-cm ground sampling distance (GSD) was acquired from the flight mission. Repeated adjacent photographs were kept for at least 85 % overlap and 45 %

sidelap. The data sets, including orthomosaic images, DSM, and true 3D model, were generated and processed using ContextCapture and Pix4Dmapper with a grid spacing of 8.5 cm. Fig. 2a denotes the region of data acquisition by the UAS. The green and red points mark the waypoints of the flight mission and all photographs acquired, respectively. Figs. 2b and 2c demonstrate the 3D scene of the study area simulated using the true 3D model, where the crown area of one landslide was modified by human infrastructure.

On the other hand, the airborne LiDAR is an excellent tool for extracting the meter-scale surface features of landslides. Consequently, quantitative studies have been successfully conducted for delineating landslides; determining risks (Gritzner et al, 2001), landslide morphology (Glenn et al., 2006; Staley et al., 2006; Santangelo et al., 2015b), and activity (Chen et al., 2005; Chang et al., 2006); estimate cut-and-fill volumes (Chang et al., 2005, Chen et al., 2006; Chan et al., 2012; Hsieh et al., 2016); and perform geological mapping (Yeh et al., 2014). In this study, an Optech ALTM3070

apparatus was used to record the positions of the laser reflecting points with a rate of up to 71,000 pulses per second when we conducted the Central Geological Survey project in 2005. In brief, the DSM is developed based on the first

reflection of the laser scanning from the sorted point clouds that are reflected from vegetation or the tops of buildings. Consequently, the last reflection, occurring after the vegetation is penetrated, records the appearance of the ground surface. Processing the nearby reflecting signals can also eliminate buildings. Thus, the ground surface without vegetation and buildings, referred to as a digital terrain model (DTM), can be obtained.

In this study, the airborne LiDAR DSMs and DTMs were generated with a grid size of 2 m from the classified point clouds. The average densities of the point clouds for DSMs and DTMs were 3.63 and 0.75 points/m$^2$, respectively. Compared with data measured using a differential global positioning system, the errors may vary according to differences in the terrains. For the study area, the mean errors, RMS, and standard deviation were 0.097, 0.126, and 0.081, respectively.

Before the morphological and tectonic analysis, the quality of the entire data set (as shown in Fig. 2a) must be evaluated. Therefore, 20 ground control points used in this study were extracted from the airborne LiDAR data set. The quality of the DSM was approximately 5 cm with a maximum error of 35 cm in the root-mean-square (RMS) deviation compared with that of the airborne LiDAR data for the open bare ground area, where the elevation of the DTM corresponds to that of the DSM, and there is neither vegetation nor buildings. Figs. 2 and 3 provide some examples of the

quality of both DEM and orthomosaic images of the study area. The morphological features of the ground surface based on the acquired DEM and photographs were analyzed and accordingly interpreted.

      The UAS images, which generate 8.5 cm pixel resolution in both the orthomosaic photo and DSM, distinguish clearly the ground and non-ground features, such as buildings and sparse vegetation. Moreover, this information is helpful at improving the airborne LiDAR data processing and point clouds classification.

Fig. 3 presents images of the same area close to the southwestern CSL slide crowns obtained from different sources and highlights the resolution and application of different data sets. In the study area, two different landforms can be readily distinguished, i.e., dense forest and sparse vegetation region resulted by human and agricultural development. The gentle slope and fractured volcanic rock mass provided a suitable environment for agriculture and habitation. The ground appearances obtained from the IKONOS satellite image (Fig. 3g; 1-m resolution), aerial photographs (Fig. 3h;

25-cm resolution), and UAS images (Fig. 3i; 8.5-cm resolution) show different levels of detail. Figs. 3g, 3h, and 3i show a dense forest covering the upper left corner of the images. Because of an annual rainfall of more than 2500 mm in this area, a vast portion of the study area is covered by vegetation. Dense vegetation often partially conceals morphological features in humid areas and has prevented detailed studies in the past on surface processes, e.g., incision or erosion.

      Figs. 3a and 3b show the DSMs and DTMs illustrating the differences in ground features with and without

vegetation and buildings. The point clouds from which the DSM and DTM profiles were derived are shown and compared in Fig. 3d. Fig. 3c indicates human and agricultural activities in the crown area of the landslide, as shown in

the lower halves of the images. Figs. 3c, 3d, 3e and 3f demonstrate the two landform regions with different vegetation coverage. The landform region with sparse vegetation corresponds and is almost equal to the region of landslide. The UAS DSM generated in this study is very similar to so-called DTM, because the terrain is not concealed by the forest canopy. Thus, the geomorphologic analysis outside the landslide region depends mainly on the airborne LiDAR DSM and DTM in our study. Overall, the UAS and airborne LiDAR datasets can be mutually compensated for the geomorphological analysis in this study.

## 4 Results of morphological analysis

Morphological analysis is a key means of landslide identification. Satellite images or aerial photographs are traditionally used to perform morphological analysis. A DEM may provide more landform information and is typically developed by pairing satellite images or aerial photographs. However, the resolution of DEMs derived either from satellite images or aerial photographs is insufficient for performing highly detailed morphological analysis. For example, dense vegetation in humid areas often conceals part of the landslide's morphological features; this situation is readily resolved by using airborne LiDAR data and high-resolution UAS images. LiDAR laser pulse emission can penetrate the barriers of different media (trees, clouds, and vegetation) based on corresponding reflections. UAS images and associated DSMs can produce data sets with decimeter to centimeter resolutions, which enable detailed photointerpretation and geological structure analysis. In this study, the DEM data acquired from the LiDAR images and UAS data set were selected as the fundamental sources for detailed morphological analysis. Fig.3 demonstrates the differences in quality and in spatial resolution.

A detailed regional geologic map was reproduced by investigation from the high-resolution DEMs prior to the landslide study, as illustrated in Fig. 4. The geological structures, e.g. lineament, fault, fold and especially the landslides, are interpreted directly from LiDAR DTM and DSM, and UAS DSM and 3D model, then validated in the field. The color patches represent strata boundaries initially from existing geologic maps, and they were then modified with LiDAR and UAS data (Yeh et al., 2014 and 2017), and again validated in the field if possible. The components contain faults, lineaments and landslides shown by scarps (Fig. 4). For the first appearance, there many landslides occurred in the study area and in the Tatun Volcano region. Comparing the size, distribution and classification, data accessibility especially for UAS flight mission, the two largest landslides (XSL and CSL) were thus chosen as the target for this study.

## 4.1 Geomorphological characteristics

The geomorphological characteristics of a landslide can be divided into the depletion and accumulation areas. Most accumulation areas are prone to erosion by subsequent surface processes because of fractured and fragmented weak debris. Contrastingly, depletion areas composed of more competent bedrock tend to remain stable and are readily detectable. Therefore, the morphological analysis in this study focuses on the depleted areas shown in the UAS and LiDAR data. Despite some small landslides in the study area being identified from satellite images or aerial photograph analysis, limited resolution and dense vegetation obscured their detailed geomorphic characteristics. The small scarps and depletion depths indicate that they are shallow landslides. Fig. 5 illustrates some of these landslides; among them, two large-scale landslides, the XSL and the CSL, were selected for study.

The two main landslides have preserved some common morphological features. In contrast to the XSL, the CSL has preserved relatively more of the original topographic features characteristic of a landslide, including the circular crown, main scarp, and transverse ridge, particularly in its southwestern flank. Fig. 6 shows the hillshade elevation view of the XSL and CSL. The CSL has many small, elongated bumps, known as transverse ridges, that indicate where displaced blocks used to rest, as well as the locations of induced ground deformation. The CSL has two remarkable shutter ridges with typical triangular facets (Figs. 6). Fig. 7 explicitly illustrates the typical landslide landforms, e.g. main scarp, shutter ridges, etc., from different perspective views. This phenomenon reveals that the sliding process has cut off two linear cliff remnants and generated these two triangular facets.

Furthermore, in Fig. 7b two linear extensional cracks right behind the main scarp represent the landslide retrogressive enlargement. The lineaments parallel to the triangular facet support the argument of landslide process, especially illustrated on Figs. 7i and 7j. The identifications of lineament are interpreted from LiDAR DTM, and the UAS DSM is used to support that the terrain is not affected by other artifacts, simply from much higher resolution 3D terrain data. The resolution of decimetric 3D terrain model illustrated as in Figs. 7g and 7h providing supplemental ground truth.

The morphological features of landslides may disappear during subsequent surface processes, most likely because of progressive landslide, surface erosion, normal faulting, and fluvial development. Consequent erosion by heavy precipitation induced by the annual northeast monsoon markedly modifies the accumulation areas of landslides, rendering them unable to retain their original topography. Moreover, the erosion may be further enhanced by normal faults cutting the toe of the slope. The trace of the JSF (also known as the SCF) is located on the southeastern side of the landslide and is shown in Fig. 5. These major factors have resulted in stream initiation and the development of drainage systems. In the depletion zones of both XSL and CSL, two gullies, marked as Xishi Creek and Chinshui Creek in Fig. 6, developed on a gently dipping slope at an angle of 6°–7°. On the XSL, the drainage basin of Xishi Creek showed a

symmetric dendritic pattern in the slide area. However, on the CSL, the drainage pattern was not dendritic but rather a curved parallel system, suggesting subsequent morphological processes.

As mentioned and illustrated in Figs. 5 and 6, the CSL is marked with circular crown, main scarp, circular concentric transverse ridges in the rear of the main body, whereas, most of the landslide morphologic components in the

XSL have been modified by human activities, e.g. the graveyard illustrated in Figs. 3b, 3c and 6. For example, the crown area of the landslide has been developed into a graveyard with clearly preserved lateral franks (Figs. 6 and 7). According to the criteria of landslide classification proposed by Varnes (1978), the two major landslides, from the currently observed landslide geomorphologic components, suggest that the landslides are best classified as rotational or translation slides.

**4.2 Estimation of the landslide dimensions**

This study considered the nature of a volcanic dome for reconstructing its morphology before a landslide. The undisturbed shape of the volcanic edifice, particularly for a stratovolcano with one main eruption center, typically mimics a conical dome. Subsequent large-scale landslide events generate morphological depressions and change the original topography. To reconstruct the paleolandform, a large depleted area may be estimated by refilling the depletion. For

example, for the case shown in the inset of Fig. 8, despite the radial shallow erosion gully, two-thirds of the northwestern flank remained conical in shape. Thus, the geometry of the northwestern flank was used to model the undisturbed conical shape of the volcano. The depleted area was manually filled by inserting several linearly aligned points within the slide area. The linearly aligned points in Fig. 9a show the undisturbed topography before a landslide. A local high envelope surface was subsequently calculated with a search circle. Because the calculated envelope surfaces may not cover all

areas, the search circle was then moved gradually around the grids to assign new heights. The procedure of determining the local high envelope surface and assigning new heights to the grids is iterative until the results are converged.

Fig. 8 shows the differences between the real and reconstructed DTMs, as shown by several profiles from the Tatun volcano group. The locations of the profile lines were selected according to the landslide area. The trend of the profile lines was selected from around the volcano summit to the toe of the slope. For every profile, the point locations were

selected from the highest local envelope surface to avoid as many points of surficial erosion as possible. The profiles before and after reconstruction were fitted by exponential and power-law functions, as shown in Table 1. The correlation coefficient values ($R$) for the profiles throughout the sliding area were calculated. High $R$ values indicate better fit to the filled DTM profile, thus correlating with the ideal conical volcano shape. The $R$ values of the reconstructed DTMs for the landslides were considerably similar to those of profiles where no landslides occurred, as indicated by the profiles P1–P3

in Fig. 8. Thus, these reconstructed DTMs for the landslides were reasonably restored to their prelandslide morphologies.

Fig. 10 shows the estimated isopach map of the cut-and-fill depth for the two main landslides. The landslide volume was calculated from elevation changes before and after each landslide. The cut-and-fill depth was estimated based on the elevation differences between the current and reconstructed DTMs. The cut-and-fill volume is based only on the difference of DTM, and did not account for some of the remaining debris on the slip surface. On the other hand, the volume does not consider how many landslide events have occurred to induce such volume due to insufficient evidence. The estimation of landslide volume indicates the minimum volume in this study. Finally, the total cut volume of the CSL and XSL was calculated to be at least $820 \pm 20 \times 10^6$ m$^3$, covering an area of approximately 7 km$^2$; this volume is approximately six times larger than that of the largest landslide ever reported in Taiwan (Chen et al., 2005 and 2006).

## 4.3 Morphological reconstruction and evolution in sliding area

The evolution of slope degradation is the most prominent type of morphological change but may not be sufficiently rapid in showing changes. The fluvial incision process may record the efficiency of the slope evolution (Chen et al., 2006). Fluvial development is invariably accompanied by riverbed incision. The gullies develop into surface runoff flow along small ditches on the bare ground; the incision of the ground surface by these ditches is determined accordingly. Therefore, to eliminate the small-scale surficial disturbances after landslides, gullies should be filled to reconstruct the original topography, as shown in Fig. 9b. The mean level of the original ground surface and average datum drawn from an enveloping surface were used to accurately fill the gullies. To obtain the trend of the topographic surface, the current DTM grid data were processed and weighted by a Gaussian function over a small area, defined by a search radius (D1 and D2, Fig. 9b). The different radii in the search circle represent different thresholds for river development and surface roundness. Within the search circle, the means of the grid data were assigned to the corresponding value applicable to the center of the circle. The search circle was then scanned over the grids. By using this method, the local unevenness induced by small gully incisions can be smoothed.

The differences between the current DTM and its enveloping surfaces can serve as an index for small-scale landslides (e.g., surficial erosion or shallow landslides). Therefore, the average datum should neither be in the uppermost nor the lowermost enveloping surface but instead be at the intermediate level, as shown in Fig. 9b. On identifying the average datum, the subsequent river incision was revealed by the shaded area, as shown in Fig. 9b. Unlike the part of the ground where gullies developed, the other portion of the landslide retained its original surficial roughness. The current relief above the averaged datum was not modified because it was deemed to because of the unevenness or roughness of the original topography, and gully erosion was not concentrated in this area. Therefore, when the calculated mean elevation exceeds the elevation of the circle center, the center elevation is assigned as the mean elevation. The reconstructed morphology is the thick line shown in Fig. 9b, and the shaded area, which is the difference between the old

and current DTM, reveals the possible areas of river incision. The hatched areas indicated the eroded volume that formed the current gully (Fig. 9b) under no pronounced morphological changes, such as a large-scale landslide. This method could effectively restore the disturbances induced by small gullies or riverbed incision. Furthermore, the different search radii may render different envelopes. The variations in the envelopes obtained from different search radii inherently reflected the range of possible errors in determining the incision depth (Fig. 9b). Fig. 11 demonstrates the upper and lower bounds of several cross-sections along the two creeks located within the slide area. The discrepancies between various reconstructed DTMs result from the different thresholds (search radii) used during reconstruction.

Fig. 12 shows the longitudinal profiles of the two creeks located on the XSL and CSL (Fig. 10). The inclined solid lines in Fig. 12 represent the current elevations along the two creeks; the dashed lines indicate the upper and lower bounds of the enveloping surface obtained by different search radii D1 and D2, as indicated in Fig. 9b. The horizontal distances were individually normalized by the analyzed river segments from upstream Chinshui Creek to downstream Xishi Creek. The analyzed segments of the river profile were completely contained within the landslide areas, from the main scarp to the junction of the pediment and the delta fan. The elevation differences between the current DTMs and envelope surface were calculated (shaded bands, Fig. 12). Moreover, the elevation differences between the upper and lower bounds denote the range of error in estimating riverbed incision.

A mean level curve represents the main trend of the incision depth distribution. If the raw data of the mean depths are used, the derived curve would appear jagged. Therefore, to obtain a smooth mean value curve, the following procedures were developed.

1) The mean incision depth should be located between the upper and lower bounds along the creeks, as calculated using the search radii D2 and D1, respectively.

2) Three gully erosion depth-versus-distance curves—the maximum, minimum, and mobile averages—are calculated from the mean of the maximum and minimum enveloping surfaces. At each location along the creek, the mean depth of the mobile average curve located with a horizontal span of approximately 400 m is calculated. The derived mean depth is then assigned as the representative mean depth at this location.

Using these procedures, a smooth mean incision depth versus the horizontal distance curve can be obtained, as shown by the bold solid lines in Fig. 12. This curve represents the possible distribution of incision depth along the creek. The results revealed a maximum valley depth of approximately 15 m, which was notably situated between the middle and lower parts of the hill slope. This phenomenon differs from common river profiles, where the deepest depth of the valley is often located near the upstream rather than the downstream portion.

## 5 Discussion

### 5.1 Landslide generation

Several processes can weaken the volcanic edifice, particularly hydrothermal activities (de Vries, Kerle and Petley, 2000; Reid, Sisson and Brien 2001; Reid, 2004; Norini and Lagmay 2005), which result in high pore pressures and enhance the alteration process, thus converting strong rocks into clays. Catastrophic collapses occur when a volcano becomes structurally unable to support its own load. These processes generate radial and concentric instabilities in case of no other structural anisotropy. Typically, these landslides propagate outward from the volcanic cone and extend downslope. However, the XSL and CSL slid perpendicular to the conical shape of the volcanic slope, thus prompting us to discuss several specific points.

In a normal situation, landslides frequently have a circular crown and main scarp. However, Fig. 10 shows an irregular scarp, and the estimated distribution of the sliding depth of the CSL showed an irregular form, marked as zone C in Fig. 10. Zone C implies a secondary slide with a circular crown after the main sliding event. Moreover, because zone A maintains a less disturbed original landslide morphology than does zone B, zone A could be the more recent of the two events (Fig. 10). Meanwhile, the twist of a parallel drainage pattern (Figs. 5, 6) and a concentric river developed within the slide area (Fig. 10) suggested that the initial slide mass experienced subsequent movement. These observations suggest more than one sliding event. Thus, the CSL is a combination of one main landslide and more than two subsequent small landslides within it. However, from the geomorphologic features denoted in Fig. 10 (Zones A, B and C), the regions show different degrees of preservation of the landslide geomorphologic components. The CSL can be interpreted to have occurred from a combination of multiple landslide events. In addition, the CSL and XSL preserved different degrees of landslide geomorphologic components, and the creeks as illustrated in Figs. 5, 6 and 11 developed within the depletion zone with different drainage patterns and varying incision depths. These observations suggest that more than one sliding event has occurred in the study area.

Finally, a comparison of the XSL and CSL revealed that typical landslide morphological features are more well-preserved in the CSL, indicating that it is newer than the XSL, assuming that the weathering rate for both landslides is similar. A large landslide volume coincides with a large accumulation area near the toe. However, as shown in Figs. 5, 6, and 11, a flat alluvial plain was observed instead of an accumulation of colluvial or talus deposits. Normal faulting along the JSF, also also known as the SCF, particularly where the trace of the JSF was reactivated as a normal fault, ultimately yielded a half graben near the toe area and thus accommodated the volume. This process may plausibly account for the missing colluvial deposits, as illustrated in Figs. 5 and 6. An additional line of evidence supporting this explanation is that the river incision profiles had maximum erosion areas located from the middle to lower slope and were abruptly

truncated near the toe area (Fig. 12, inset). This particular landform indicates the subsiding process along the fault trace, consuming the deposited debris that would otherwise be present at the slope toe. The remaining displaced material in the CSL suggests a combination of multiple landslide events. However, most of the displaced material in the XSL has been eroded away and it is not possible to estimate how many events are involved in the XSL. On the other hand, the two

linear extensional cracks right behind and parallel to the main scarp represent the landslide retrogressive enlargement (Figs. 7b, 7i and 7j). These phenomena denote the active and progressive evolution of the CSL.

To conclude the landslide generation, the normal faulting in the region started from 400 Ka and is activated continuously ever since. The faulting was identified in the Taipei basin area and northeastern offshore Taiwan, with the fault line situated on both sides of the study area (Figs. 4 and 5). And the fault line was recently identified and linked

together as only one normal fault in Tatun Volcano region. In conclusion, for the relative age of the landslide, we interpret that the landslide has been triggered since right after normal faulting started and the formation of Tatun Volcano, which is far later than 200 Ka. Regarding to the different generation of landslide, the geomorphologic components also show different degrees of preservation within the two observed landslides. Furthermore, the CSL is interpreted to have occurred from a combination of multiple landslide events.

**5.2 Landslide slip surface and volume estimation**

According to geological data and field observations, the Tatun volcanic rocks overlie the Mio-Pliocene sedimentary rocks (Figs. 1 and 4). Because of the clear contrast in the rock strength and strata unconformity, the contact surface of the two rock types may easily serve as the rupture site for the sliding surface, thus indicating that most landslide debris were eroded or slid away when the contact surface was largely exposed. Thus, although the estimated volume was considered

the minimum landslide volume, volume estimation shall approximate the actual volume in this special case.

Regarding the landslide volume, the position and morphology of the slip surface indeed will affect the calculated cut-and-fill volume. In this study the slip surface is difficult to observe in the field due to soil cover and has not been definitely identified. Nevertheless, the sedimentary rock basement and the volcanic rock cover have been well mapped both on the geologic map and in field survey in the region (Figs. 1 and 4). Based on the distribution of rock types, it is

supposed that the contact between the volcanic cover and the underneath sedimentary rocks may serve as a weak plane for the slip surface. On the other hand, the calculated landslide volume is derived from the difference of DTM, which denotes only the minimum volume, and does not take into account the remaining debris still resting on the supposed slip surface, especially for the larger landslide CSL, as shown in Fig. 10.

For the XSL, the maximum cut depth was approximately 150 m. The maximum cut area was situated in the central

zone of the sliding area and showed a symmetrical reverse-conic shape; a uniform erosion process in the accumulated

area may account for this pattern. For the CSL, the sliding mass had a sliding depth of approximately 200 m and a wide and flat bottom. The volume of the CSL landslide was approximately three times larger than that of the XSL. It is noted that there are two small creeks, the Xishi and the Chinshui, and several incised gullies of the ground surface (Figs. 6, 7 and 10). The Fig. 12 illustrate the average incision depth is ranging few to some 15 meters, however limited closed to the creeks and gullies. Comparing with the overall landslide region of 7 $Km^2$ and over one hundred meters in depth, the effect of gully incision affecting the landslide volume estimation may be neglected. Nevertheless, these surface processes may affect and accelerate the landform modification within landslide area, and play as a key role in the geomorphologic evolution.

Although the assumed ideal volcanic conical dome may deviate from the true shape of the topography, the estimated results provided useful information about ideal magnitudes of the scale and volumes of landslides, which are several times higher than the magnitudes previously reported in Taiwan.

### 5.3 Landslide evolution

Surface deformation (e.g., normal faulting) may enhance and accelerate surface processes (Densmore, 1997; Strecker and Marret, 1999; Ambrosi and Crosta, 2006; Bucci et al., 2013 and 2016; Scheingross et al., 2013). The study area morphology revealed that these processes extended from the deformation zone toward other parts of the landscape. The evolution of the riverbed profile over a short time scale clearly demonstrated such processes. As shown in Fig. 12, the substantial subsidence of the hanging wall led to a sudden decrease in riverbed elevation at the boundary of the footwall and hanging wall, particularly on the footwall side. The subsiding normal fault enabled faster incision on the footwall at the section where the river intersected the fault. Meanwhile, the sharp fault scarp could be smoothed rapidly or even destroyed by many small-scale landslides along the footwall side of the fault trace. In brief, the largest amount of river incision was not necessarily at the location nearest to the fault scarp (marked as A in Fig. 12) but was located at a certain distance from the fault scarp (marked as B in Fig. 12). The observed maximum incision depth was approximately 15 m, implying that the actual maximum incision could be several times greater. Because washing out the colluvium and producing an incision require considerable time, the incision of tens of meters implies that the erosion processes continued long after the main landslide occurred.

In northern Taiwan, the tectonic activity of the region is in extensional regime. The Shaochiao normal faulting contributed to the formation of Taipei Basin by over one thousand meter throw of the fault separation. The normal faulting has been very well documented recently, and concludes a very active faulting behaviour (Teng et al., 2001; Shyu et al., 2005; C.T. Chen et al., 2007 and 2010; Huang et al., 2007; Chen et al., 2010). The results showed the normal faulting slip rates between 8.2-1.8 mm/yr. This normal faulting not only resulted in the formation of the Taipei basin, but

also probably related to the continuous eruption of the Tatun Volcano (Teng et al., 2001). The faulting that was identified in the Taipei Basin situated on south-western side of the study area. The faulting in northern offshore Taiwan has also been investigated and identified to be active normal faulting (Hsiao, 1997; Hsiao et al., 1998). Recently, these faulting observations were interpreted to be linked together as only one normal fault in Tatun Volcano region across the study area. The total length of the newly observed Jinshan normal fault is more than 130 Km, much longer than previous thought.

The systematic historical record of the study area dates back to the 18th century. Two major geological hazards have been documented: seismogenic normal faulting responsible for the formation of Taipei Lake in 1694 (Yu, 1959; Tsai, 1985; Shieh, 2000) and the Jinshan Tsunami triggered by the Keelung earthquake in 1867 (Tsai, 1985). However, the Jinshan landslides, especially the evolution, were correlated to these historical events. The seismogenic Taipei Lake illustrates the activity of the JSF in the most recent event. Furthermore, a recent borehole drilled in the Jinshan Delta resulted in an unconformable contact between the Oligocene Wuchihshan Formation and recent alluvial gravel deposits at 551.6 m below the surface (Lin, 2005). Borehole logs show that the gravel deposits occurred uninterrupted from the ground surface until 551.6 m, and rocks of the Wuchihshan Formation were recorded below 551.6 m. The borehole is only approximately 300 m from the inferred fault line trace (Fig. 5). This contact relationship implies an association between basin subsidence and regional normal faulting. Normal faulting suggested at least 880 m of stratigraphic separation of throw in this area, as determined using borehole log information and regional stratigraphic correlation.

Overall, the present findings, in addition to previous studies on regional geology, geophysical observations, and recent tectonic setting interpretations (Wang, 1999; Huang, 1988; Hsiao, 1997; Hsiao et al., 1998; Lin, 2005), highlight the normal faulting activities in Jinshan. Moreover, the observed morphological discontinuities suggest that normal faulting may have cut through and truncated the volcanic cone. Thus, a fault scarp was formed, which itself is being a morphological discontinuity (Fig. 13). The normal faulting process created a steeper slope and fractured the nearby rocks. The normal faulting, which cut through the volcanic and sedimentary rocks by high dipping angle, will create a steep slope at the toe region. Meanwhile, the strata discontinuity plane dipping to the same direction has a lower dip angle. The phenomena will generate a so-called "slope daylight" region (e.g., Yeh et al., 2017), and will create one unstable/hazardous region, as illustrated in Fig. 13 and shown in Fig. 12, where the maximum elevation difference situated near the toe of the slope. The continuously activated normal faulting may crate furthermore the daylight of the slope as the evidence of the exposed sediment rocks in the upslope region. The weakening of rocks, in addition to the morphological discontinuities, induced slope instability in the scarp along the fault. Furthermore, successive faulting events enlarged the sliding area and modified the form of the overlying accumulated deposits. The combined actions of the elevation differences between the scarp and landslides along the active normal fault promoted the removal of the

colluvium near the toe. As the normal faulting reactivated continuously, one half-graben, the Jinshan Delta, is thus formed allowing accumulation of fluvial and landslide deposits. The eroded material consequently formed a large alluvial delta at the outlet of the main stream (Fig. 5). The overall landform and structural evolution of the landslides are proposed and schematically shown in Fig. 13.

## 6 Conclusions

The presented UAS technology and LiDAR data reveal the relevance of high-resolution digital surface images, namely DTMs, DSMs, and true 3D models, for effective surveys of obscure old landslides in a volcanic region. To investigate the morphological evolution, procedures for paleolandform reconstruction were proposed, based on the concept of the envelope surface, according to relict topography from the Tatun volcano group in Northern Taiwan. This analysis was extended to geomorphic applications, such as estimating the cut-and-fill volume and interpreting fluvial processes. The study area in the Tatun volcano group provides the following valuable insights into the morphological interactions: 1) normal faulting may have triggered the main landslides, 2) erosion near the fault scarp enhanced the transportation of colluvium and river incision, and 3) gully incision indicated that the main landslides may have resulted from old landslide events. New tectonic activities because of normal faulting will likely result in the reactivation of the existing landslides posing life-threatening situations, particularly if the slope toe is being eroded by alluvial processes transporting sediments downstream and by graben subsidence daylighting the dip slope and reactivating landslide. The potential multiple geological hazards as anticipated by the proposed geomorphological evolution because of the fault activity are worthy of further investigation for hazard mitigation purposes.

## 7 Acknowledgments

We thank Fu-Shu Jeng of National Taiwan University for revising the first draft of this paper and Jian-Cheng Lee and Su-Ching Chou of the Institute of Earth Sciences, Academia Sinica for their constructive comments and assistance. We also thank Christopher Fong for providing technical advice. This study was partly supported by the Central Geological Survey, Ministry of Economic Affairs, Taiwan R.O.C.; Tatun thematic project from the Institute of Earth Sciences, Academia Sinica; and Ministry of Science and Technology, Taiwan R.O.C.

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

**Fig. 1: Geological and geomorphological map of the study area. (a) Geodynamic map of the island of Taiwan and its northern region, illustrating the regional tectonic structures and the geology near the sliding area. (b) Geological cross-section of the study area, considered to be a region of imbricated, west-vergent, NE–SW trending folds and thrusts in the Taiwan mountain belt. The compressive synorogenic and postorogenic activities separate the successive sedimentary rocks by the Kanchiao fault (KCF), Jinshan fault (JSF), and Shanchiao fault (SCF, also known as the Jinshan Fault through normal faulting). Two landslide sites, the Xishi landslide (XSL) and Chinshui landslide (CSL), are denoted in the figure. The corresponding strata in the cross-section, from bottom to top, are the late Oligocene Wuchihshan Formation (Wc), Miocene Mushan Formation (Ms), Miocene Taliao Formation (Tl), Miocene Shihti Formation (St), and Mio-Pliocene Kueichulin Formation (Kct).**

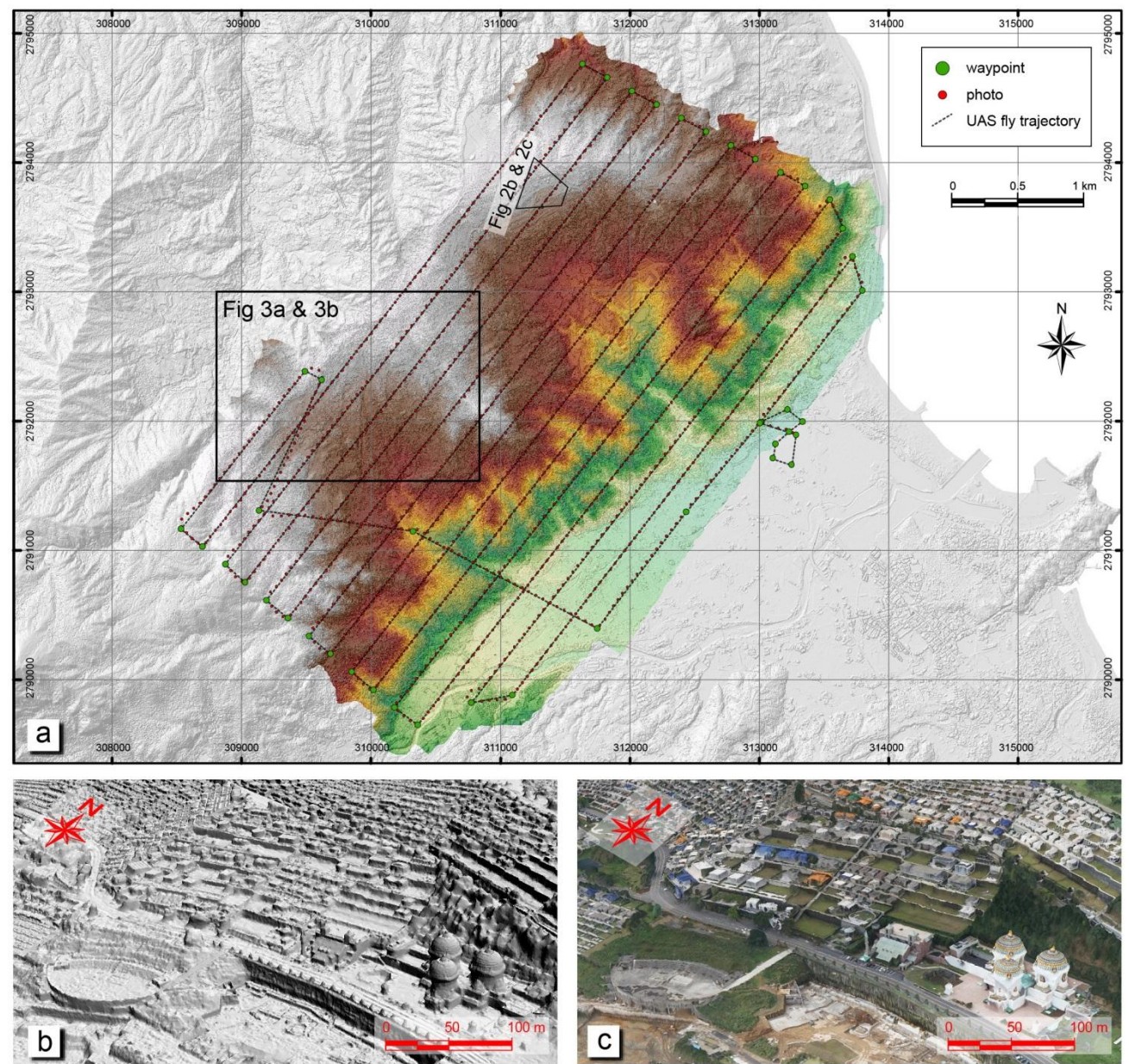

**Fig. 2: Data acquisition from an unmanned aircraft system flight mission. (a) DSM generated by the UAS for a total area of 14.7 km² with a GSD of 6.6 cm, where points and lines represent the positions and path of the flight mission, respectively, and the monotone gray area is the LiDAR DSM; (b) side view of the hillshaded true 3D model; and (c) image draped onto the true 3D model. The side views of (b) and (c) are captured at the same position and orientation.**

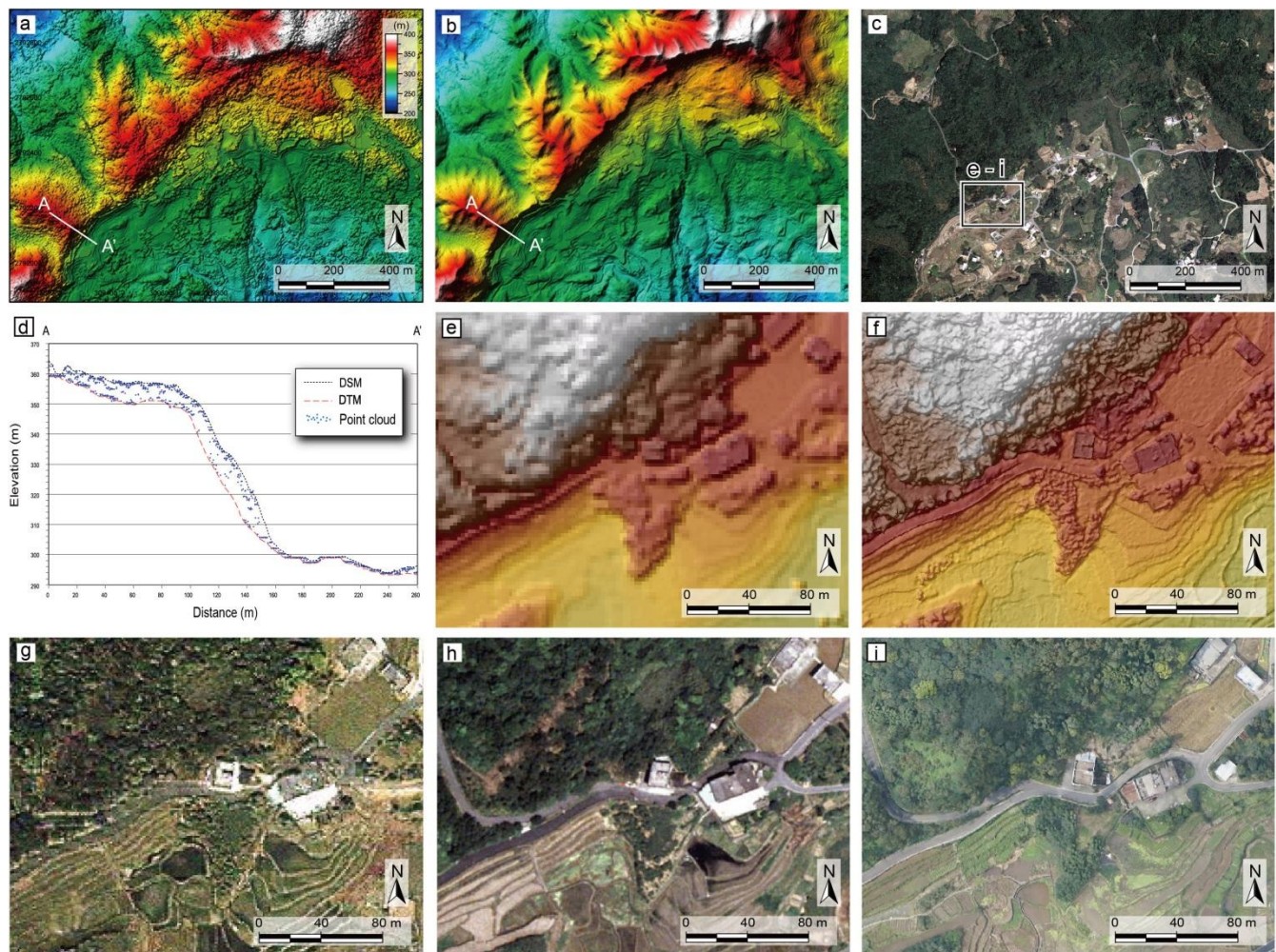

**Fig. 3: Comparison of images for the same area obtained from different sources. The area of the images is indicated in Fig. 2. (a) 2-m resolution LiDAR digital surface model (DSM); (b) 2-m resolution LiDAR digital terrain model (DTM); (c) orthorectified aerial photograph captured in March 2003, 25-cm resolution; (d) the difference between the DTM and DSM represents the height of vegetation, whereas the DTM recorded the elevation of the bare ground surface, illustrated from LiDAR point clouds. The locations of the profile A-A' marks are indicated in Figs. 3a and 3b. (e) enlargement of the 2-m resolution LiDAR DSM; (f) enlargement of the 8-cm resolution UAS DSM; (g) enlargement of the 1-m resolution IKONOS satellite image captured in November 2003; (h) enlargement of the orthorectified aerial photograph; and (i) enlargement of the orthomosaic UAS images. The enlargements are indicated in Fig. 3c.**

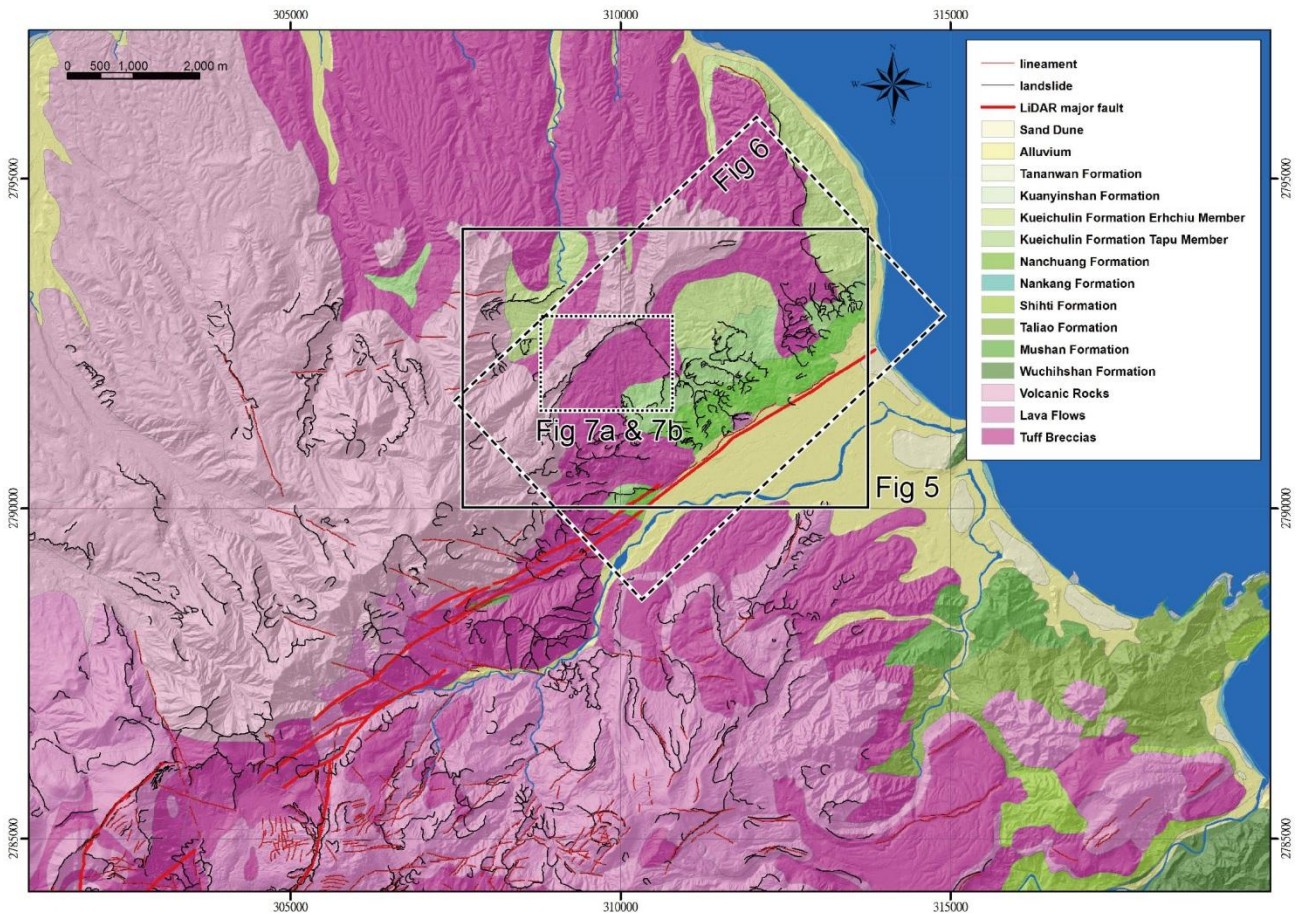

**Fig. 4: Regional geological map around the study area. Note that different components are illustrated, including faults, lineaments and landslide scarps. And many landslides identified around the Tatun Volcano region. The study area was composed of sedimentary rock and superposed by volcanic rocks. Part of the volcanic rocks were erode away so as to exposed the beneath Plio-Pleistocene sedimentary rock.**

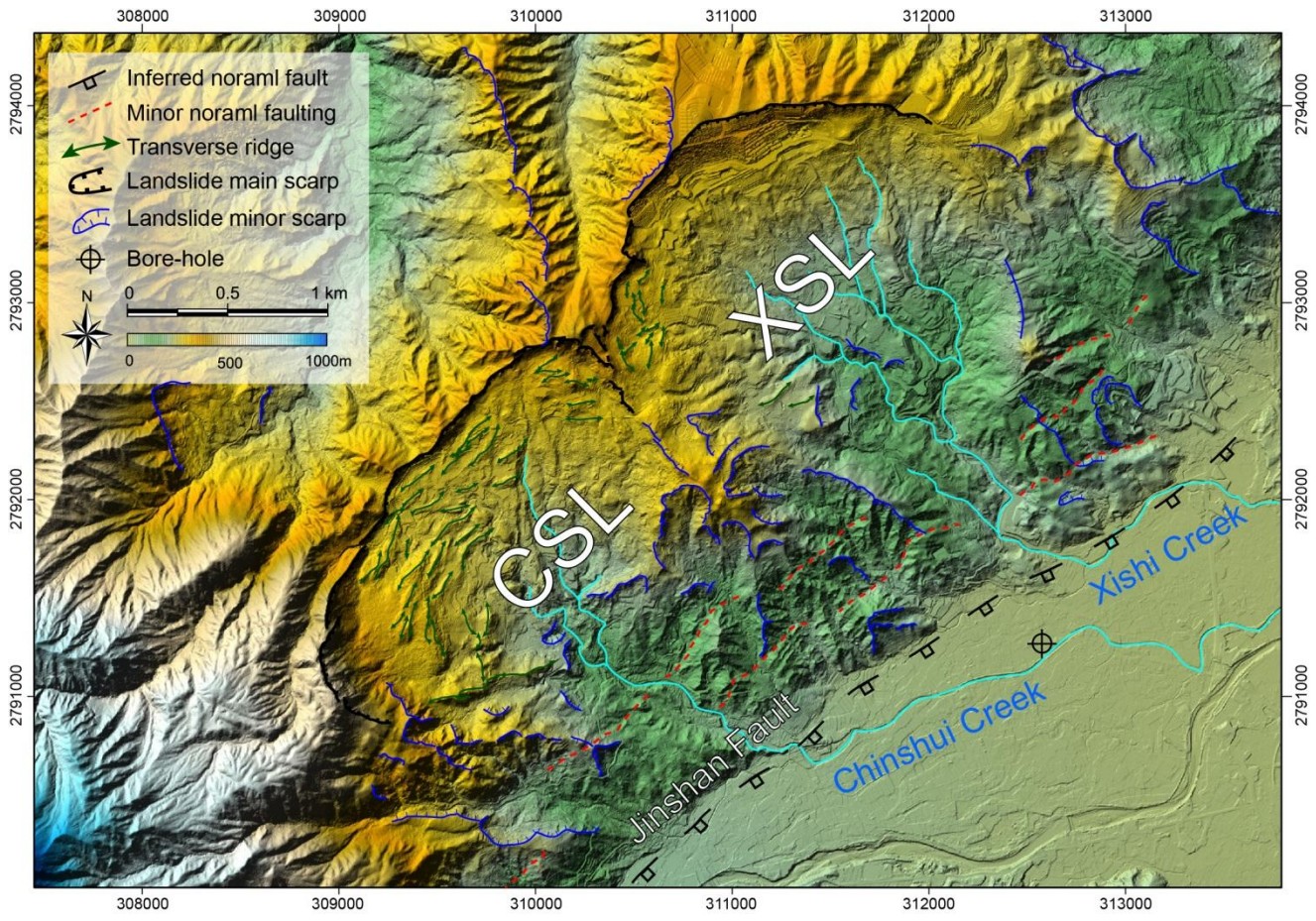

**Fig. 5: Hillshade map of the study area obtained from the LiDAR DTM. The morphological features in the depleted zone have been preserved, namely the main scarp, minor scarp, and transverse ridges. Notably, several small landslides are located on the volcano flank within the two main landslides. The region and the location of this figure are indicated in Fig. 4.**

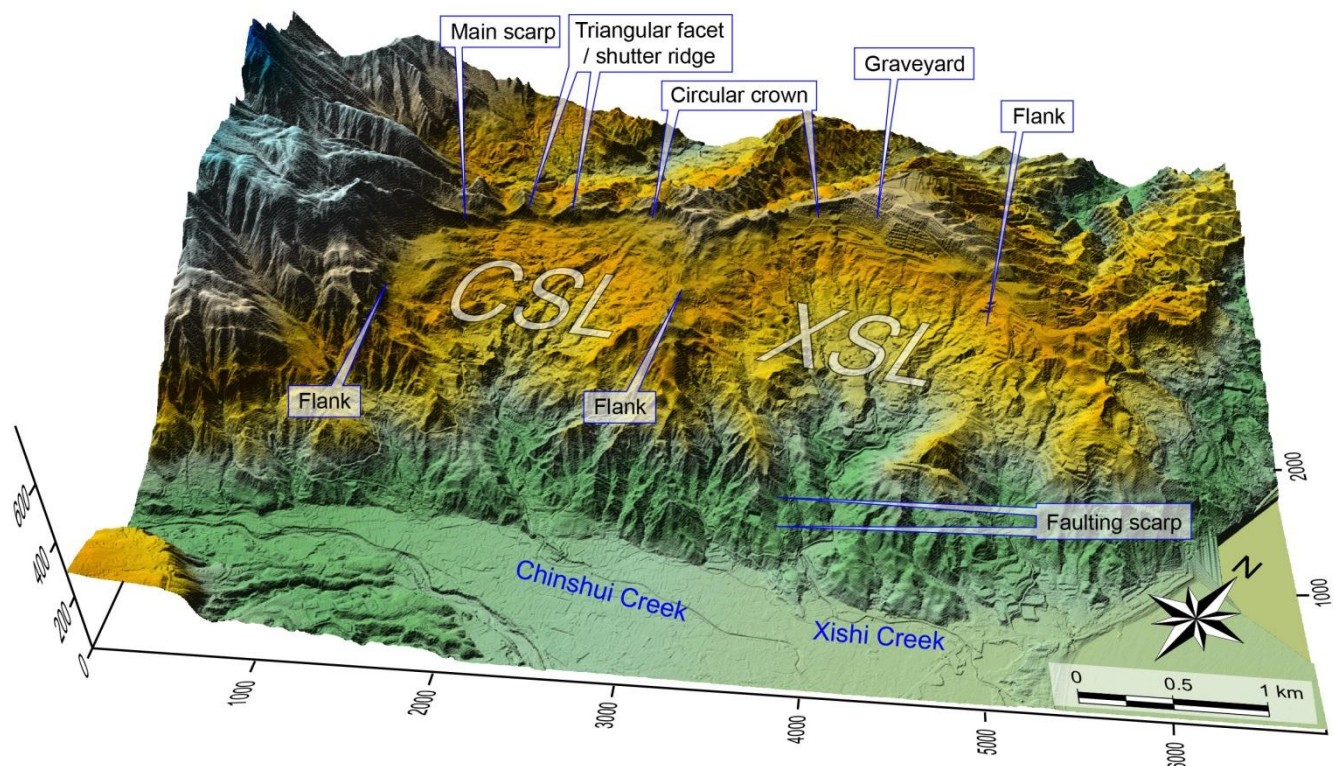

**Fig. 6: Perspective view of the 3D hillshade map of the study area from LiDAR DTM. The morphological features, namely circular crowns, flanks, main scarps, shutter ridges, and triangular facets, can be observed. However, part of the main scarp in the XSL was significantly modified as a graveyard (Figs. 3b and 3c). Only part of the depletion zone was preserved on the two main landslides. In the CSL, clearer topographic features of the sliding process were preserved, possibly indicating that the CSL is newer than the XSL. The location of this figure is indicated in Fig. 4.**

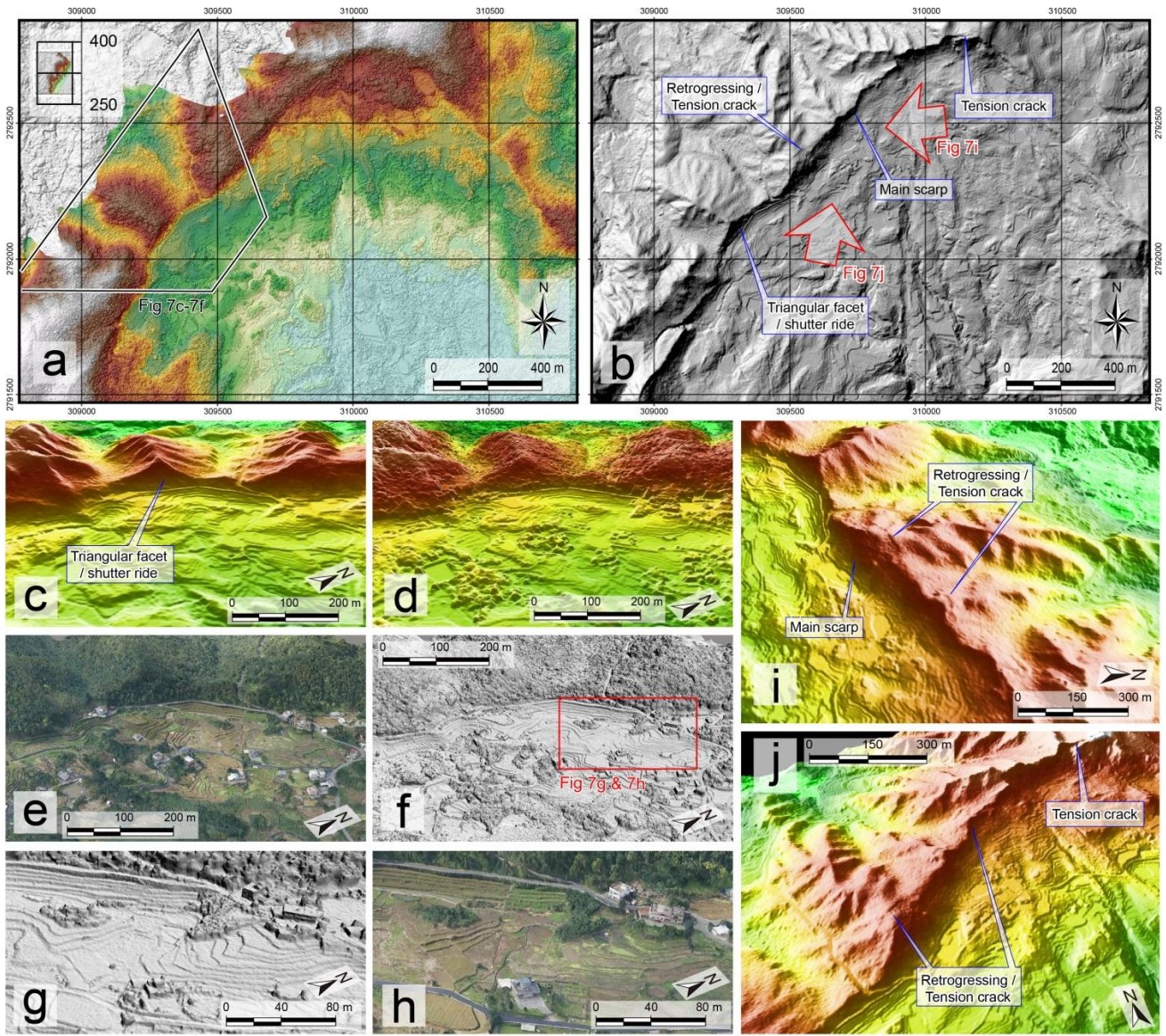

**Fig. 7: Comparison of topographic datasets for the same area acquired from different sources, including airborne LiDAR DSM, DTM, and UAS DSM. (a) 8.5 cm resolution UAS DSM data around the crown area of CSL ; (b) hillshade image from 2 m resolution LiDAR DTM, Figs. 7a and 7b denote the same region and are indicated in Fig. 4; (c) magnified side view around the main scarp area from LiDAR DTM; (d) magnified side view from LiDAR DSM; (e) UAS 3D model magnified side view with optical images; (f) UAS 3D model magnified hillshade side view, where Figs. c, d, e and f share the same perspective, indicating on Fig. 7a; (g) and (h) enlargement of the UAS 3D model, indicating on Fig. 7f; (i) color hillshade side view around main scarp; (j) another perspective view around main scarp; the direction of perspective view direction indicated on Fig. 7b.**

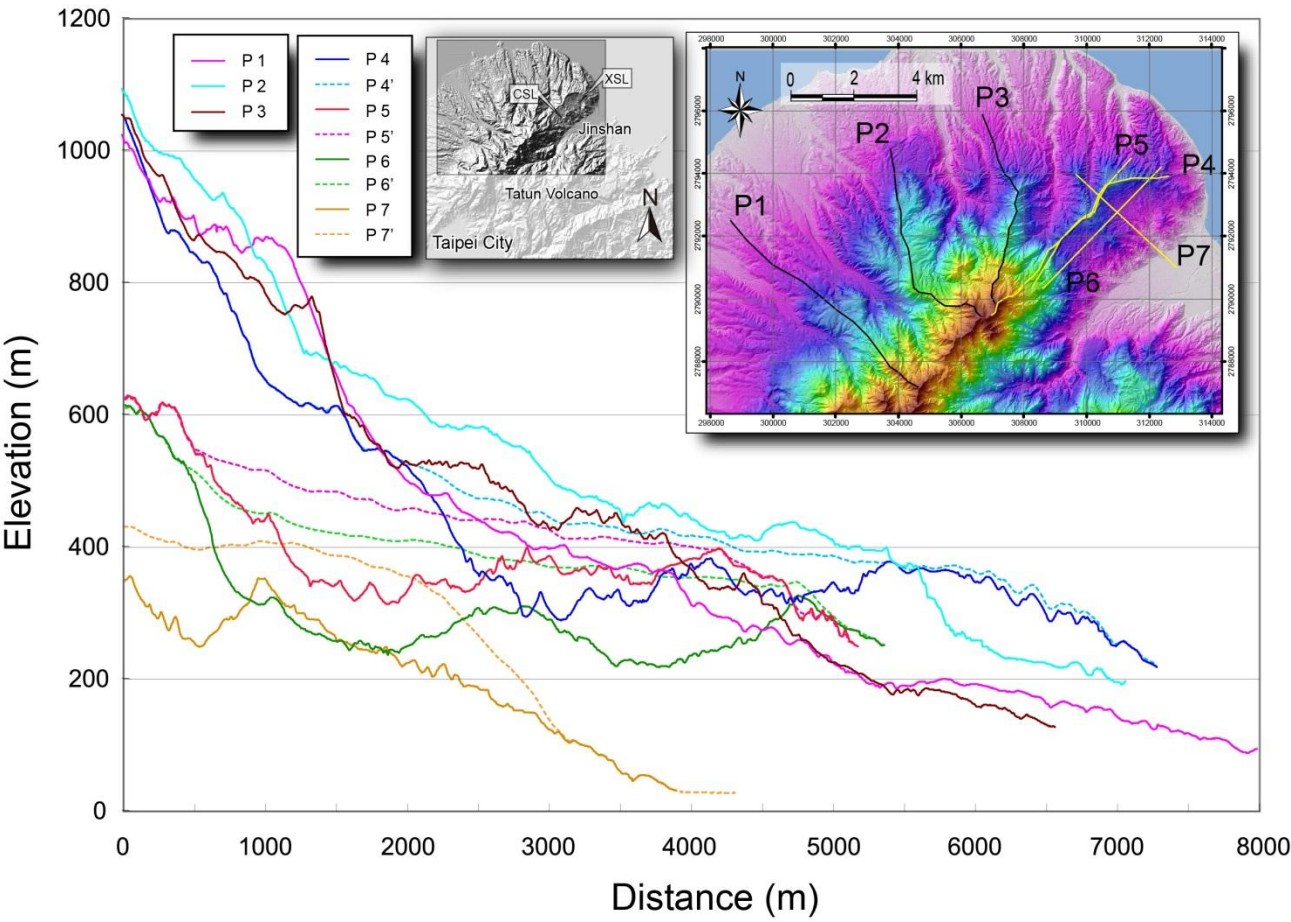

Both Figs. 7i and 7j are illustrated from LiDAR DTM. The morphological features, e.g. main scarp, shutter ridge and triangular facet, are clearly observed. Note that retrogressive tension cracks were identified above the main scarp.

5    Fig. 8: Topographic profiles of the Tatun volcanic cone. For P4, P5, and P6, profiles from the original (solid line) and the reconstructed (dashed line) DTMs are shown. The profile locations are shown in the inset in the upper right corner. The original ground surface is extracted from LiDAR DTM.

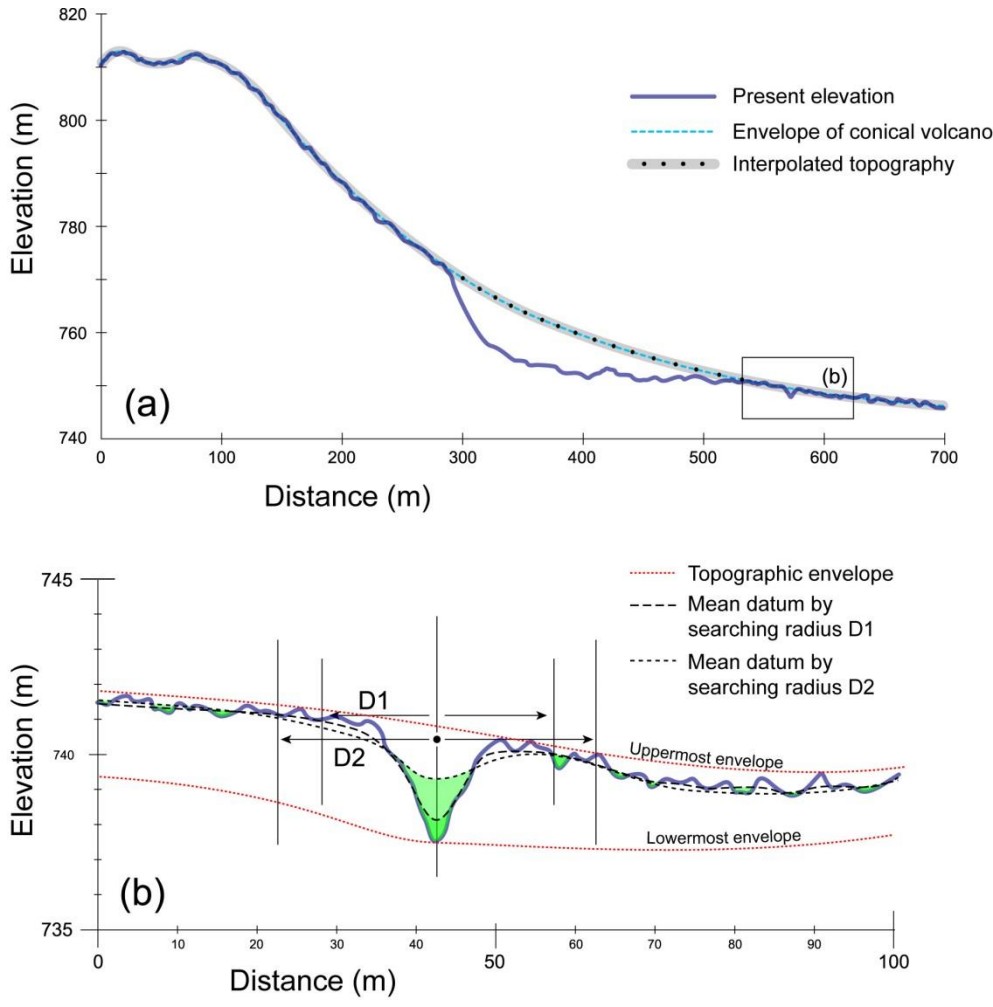

**Fig. 9: Schematic illustration showing how the envelope of paleomorphology was obtained from the current DTM morphology. (a) The configurations of the undisturbed volcanic dome and current profile. The undisturbed volcanic dome facilitates the selection of manual interpolation points that better fit the volcanic edifice. (b) The configuration of the possible mean paleodatum correspondent to the other two enveloping surfaces. The shaded area represents the river incision. D1 and D2 correspond to different radii for topographic regression and restoration. The idealized profile was simplified and modified from a part of the topographic profile P4.**

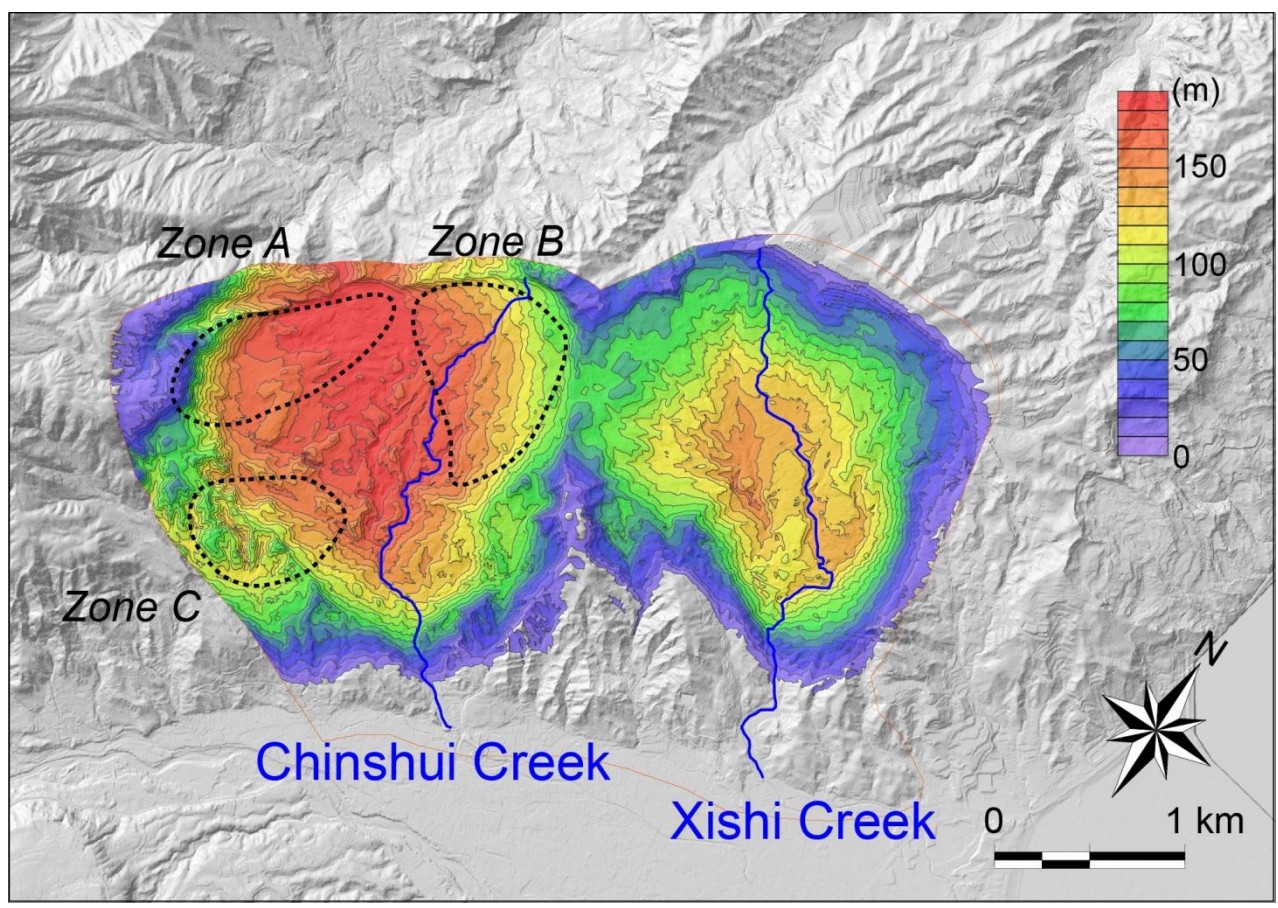

**Fig. 10: Isopach map of the landslides. The depleted landslide mass is calculated from the difference in height between the current and reconstructed topographies. The contour bands are shown at 10-m intervals. The original ground surface is extracted from LiDAR DTM.**

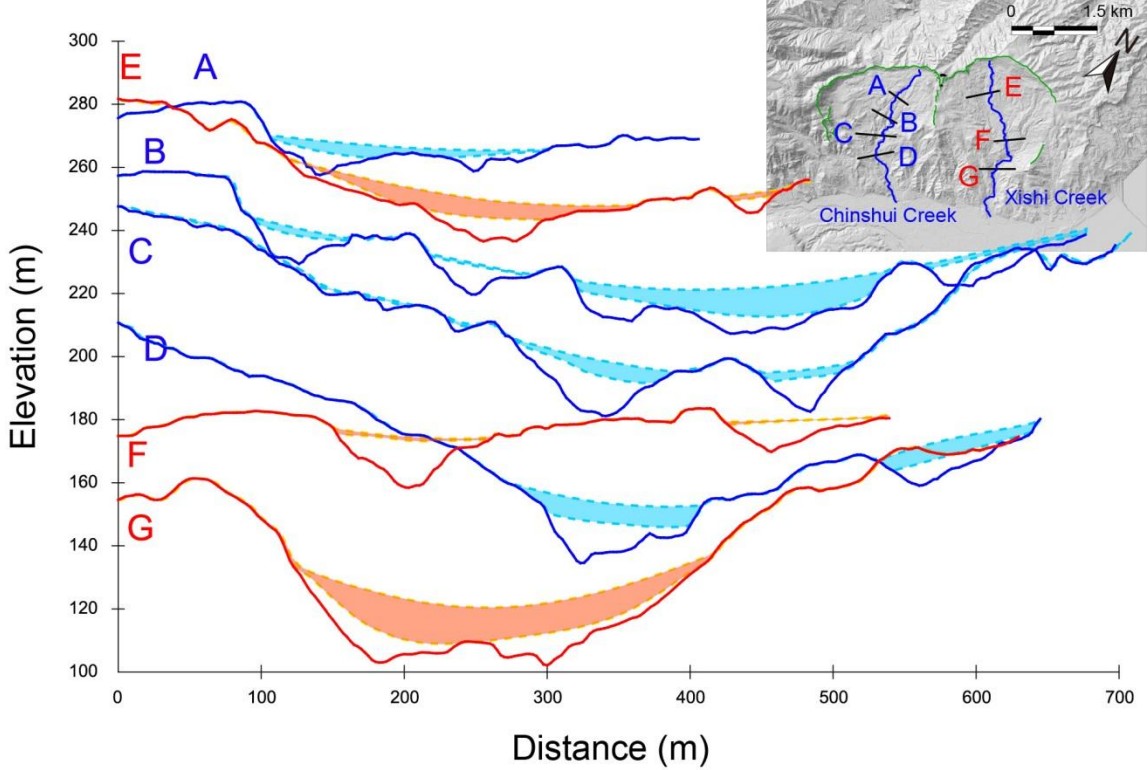

**Fig. 11: Cross-sections of the river valleys situated in the landslide area. The shaded area indicates the possible datum ranges obtained within various search radii (D1 and D2 in Fig. 9b). If the current elevation is lower than the possible paleodatum, this depth represents the amount of the incision. The upper and lower bounds are indicated by the semiopaque dashed line. The locations of the cross-sections in the landslide area are shown in the inset. The original ground surface is extracted from LiDAR DTM.**

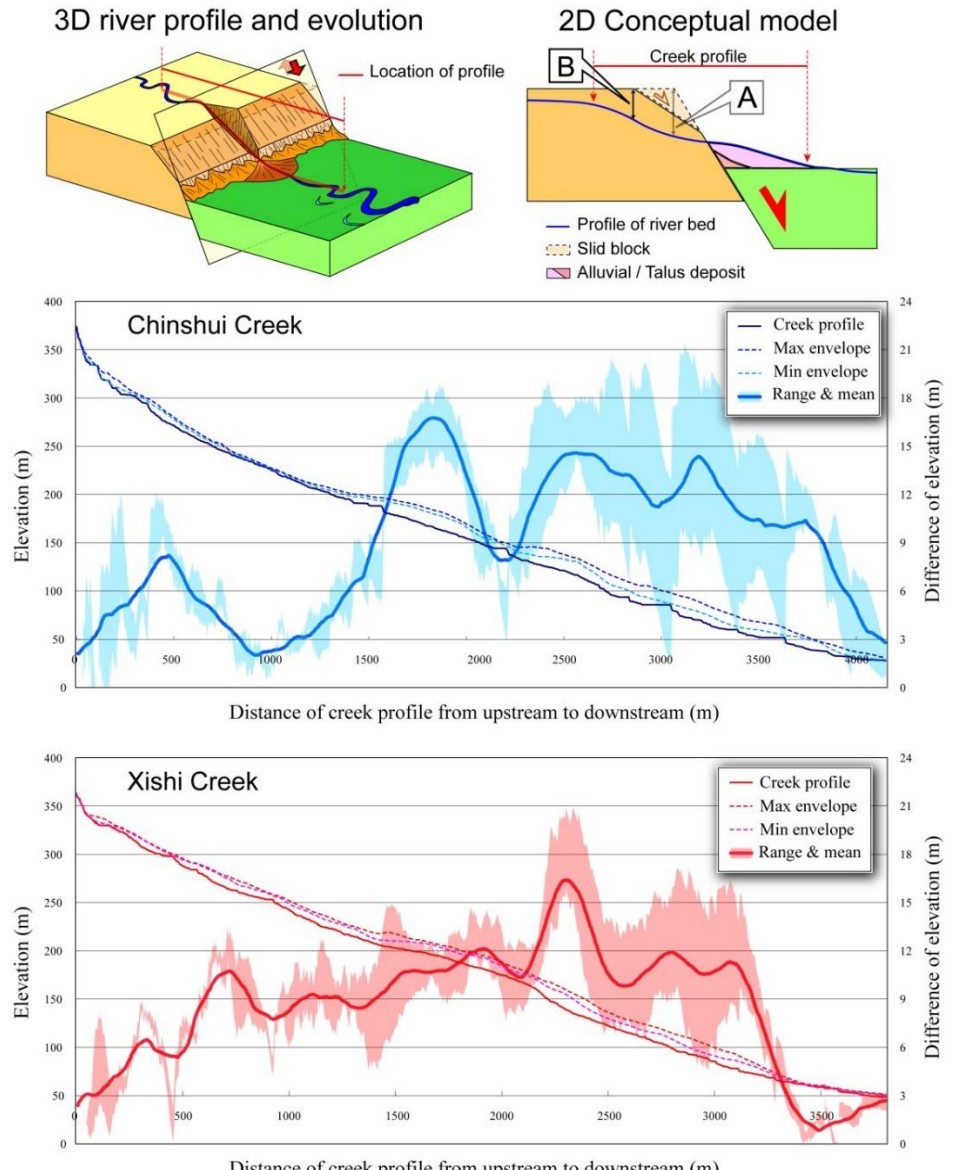

Fig. 12: River longitudinal profile and morphological evolution in the sliding area. Because of normal faulting, the substantial subsidence of the hanging wall led to a sudden reduction of the river bed. The block diagram shows the relationship between the river profile and normal faulting. The sharp fault scarp can be smoothed or even eroded by many small-scale landslides along the fault line. Therefore, the largest section of river incision may be located far from the fault scarp (marked as B), rather than close to the fault scarp (marked as A). The incision of creeks is obtained from the difference between the current and reconstructed profiles. The locations of the creek segments are shown in Fig. 11. The profile length has been normalized using the differences in the total length between the water source and basin mouth. The shaded bands form the error bar derived using different methods. The terrain elevation is extracted from LiDAR DTM.

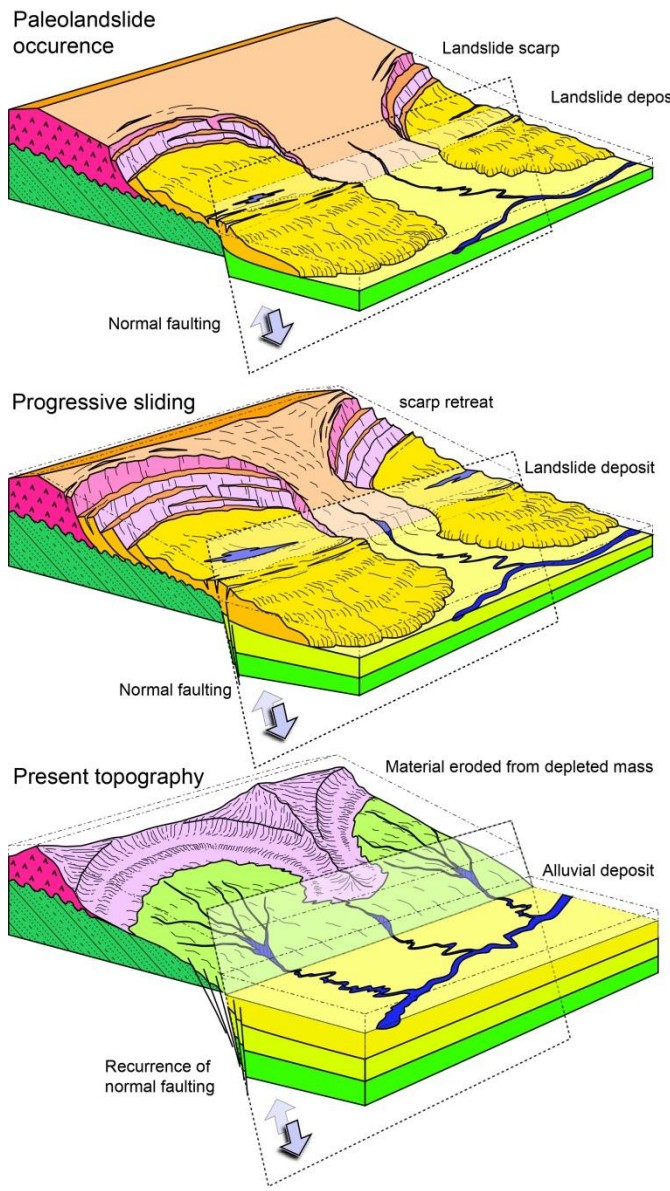

**Fig. 13: Schematic illustration of the evolution of the landslides. The triggering mechanism may be associated with normal faulting of the JSF (also known as the SCF), which transects the slope toe. The accumulation zone crossing the fault was eroded and transported, resulting in the current plain, as shown in the lower figure.**

**Table 1: Correlations of several profiles from the DTM data and curve-fit results**

| Profile | Type of curve to be fitted | | Corresponding $R^2$ | |
|---|---|---|---|---|
| | Exponential | Power | $R^2$ (exp) | $R^2$ (p) |
| P1 | $1{,}006.2 \times$ Exp (-0.0003x) | $1 \times 10^6 x^{-1.0398}$ | 0.9876 | 0.9412 |
| P2 | $1{,}020.2 \times$ Exp (-0.0002x) | $14{,}695 x^{-0.4204}$ | 0.9554 | 0.9808 |
| P3 | $1{,}075.9 \times$ Exp (-0.0003x) | $16{,}859 x^{-0.4502}$ | 0.9743 | 0.9482 |
| P4 | $830.9 \times$ Exp (-0.0002x) | $25{,}236 x^{-0.5054}$ | 0.7843 | 0.8642 |
| P4$^*$ | $908.5 \times$ Exp (-0.0002x) | $15{,}855 x^{-0.433}$ | 0.9148 | 0.9914 |
| P5 | $484.3 \times$ Exp (-9E-5x) | $1{,}133 x^{-0.146}$ | 0.4923 | 0.4167 |
| P5$^*$ | $599.5 \times$ Exp (-0.0001x) | $2{,}919 x^{-0.2484}$ | 0.8959 | 0.7297 |
| P6 | $384.7 \times$ Exp (-1E-4x) | $776 x^{-0.1321}$ | 0.3681 | 0.2944 |
| P6$^*$ | $526.3 \times$ Exp (-0.0001x) | $1{,}973 x^{-0.2109}$ | 0.8912 | 0.8307 |

*: Profiles after reconstruction; $R^2$: coefficient of determination; DTM data are fit pursuant to both exponential and power laws. The locations of the profiles are shown in the upper right corner; P1, P2, and P3 did not cross the sliding area, whereas P4, P5, and P6 crossed the sliding area. To illustrate the different effects of erosion, the original and reconstructed (marked with *) profiles are shown for comparison purposes. Because P7 (marked in Fig. 8) does not trend in the direction of the volcano slope, it was not included in the calculation.