# Peer review of "Geomorphological evolution of landslides near an active normal fault in Northern Taiwan, as revealed by LiDAR and unmanned aircraft system data"

_Natural Hazards and Earth System Sciences, 2017_

## Referee Comment (RC1) · Anonymous Referee #1 · 2 Aug 2017

nhess-2017-227 Geomorphological evolution of landslides near an active normal fault in Northern Taiwan, as revealed by LiDAR and unmanned aircraft system data

This study analyzed landslide morphological characteristics and geomorphological evolution using lidar and UAS data in northern Taiwan. The morphological reconstruction showed that the total volume of landslides reached 820 x 10ˆ6 mˆ3. This paper is interesting for the evaluation of landslide evolution and the assessment of related landslide hazards. However, the reviewer has some comments regarding landslide types, data, and methods that would need to be verified by authors.

[Figure]

1. Little information on landslide types in the study area was explained in the manuscript. Landslide types are important for discussing the landslide evolution. It would be better to show landslide types and processes analyzed in this study referring Varnes (1978) or Cruden and Varnes (1996).

2. The authors emphasized importance of UAS and lidar data. However, it was not clear how did authors use these DSMs for the geomorphological analysis, respectively. For example, the authors explained that USA had the disadvantage that the DSMs included the vegetation height. How did authors use the DSMs for the analysis? Were the geomorphological analysis and the reconstruction performed by lidar data alone?

3. The volume of the CSL was six times larger than that of the largest landslide ever reported in Taiwan which was triggered by the Chi-Chi earthquake. How did the authors assume that the CSL triggered by the single earthquake event? Additionally, the authors assumed that current topography in the CSL corresponded to the slip surface of the original landslide (Fig. 13). Did authors have geological evidences of that? Detection of the slip surface is important for estimating the volume.

---

## Author Comment (AC1) · 7 Sep 2017

Thank you for the helpful comments on our manuscript. Please find below our response and modifications that we have revised in the manuscript following the referee's comments and suggestions.

Anonymous Referee #1 This study analyzed landslide morphological characteristics and geomorphological evolution using lidar and UAS data in northern Taiwan. The morphological reconstruction showed that the total volume of landslides reached 820

x 10ˆ6 mˆ3. This paper is interesting for the evaluation of landslide evolution and the assessment of related landslide hazards. However, the reviewer has some comments regarding landslide types, data, and methods that would need to be verified by authors.

Referee #1-1. Little information on landslide types in the study area was explained in the manuscript. Landslide types are important for discussing the landslide evolution. It would be better to show landslide types and processes analyzed in this study referring Varnes (1978) or Cruden and Varnes (1996).

Response #1-1: According to the criteria of landslide classification proposed by Varnes (1978), the two major landslides analyzed in this study may be classified as rotational or translational slide. Detailed characteristics of the two slide types commonly include circular crown, main scarp, minor scarps, circular transverse ridges and lateral franks. We have newly added a paragraph in the manuscript to explain the observed landslide types in the study area. The paragraph is as follows: "As mentioned and illustrated in Figs. 5 and 6, the CSL is marked with circular crown, main scarp, circular concentric transverse ridges in the rear of the main body, whereas, most of the landslide morphologic components in the XSL have been modified by human activities. For example, the crown area of the landslide has been developed into a graveyard with clearly preserved lateral franks. According to the criteria of landslide classification proposed by Varnes (1978), the two major landslides, from the currently observed landslide geomorphologic components, suggest that the landslides are best classified as rotational or translation slides."

Referee #1-2. The authors emphasized importance of UAS and lidar data. However, it was not clear how did authors use these DSMs for the geomorphological analysis, respectively. For example, the authors explained that USA had the disadvantage that the DSMs included the vegetation height. How did authors use the DSMs for the analysis? Were the geomorphological analysis and the reconstruction performed by lidar data alone?

Response #1-2: Both the airborne LiDAR and UAS datasets are used in this study. The UAS DSM and the orthomosaic photos generated in this study are as high as 8.5 cm in pixel resolution, thus, the ground information is much more easily identified for regions of building and sparse vegetation. Especially, within the two major landslides CSL and XSL, the terrains have been affected by human and agricultural development, which is marked by a terrain with minor and low vegetation. The UAS DSM dataset is useful for processing and separating the DSM and DEM derived from airborne LiDAR dataset, because the UAS DSM is most informative at distinguishing the ground facts. In response to the comment, the above-mentioned points are added in the revised manuscript as follows: "The UAS images, which generate 8.5 cm pixel resolution in both the orthomosaic photo and DSM, distinguish clearly the ground and non-ground features, such as buildings and sparse vegetation. Moreover, this information is helpful at improving the airborne LiDAR data processing and point clouds classification. In the study area, two different landforms can be readily distinguished, i.e., dense forest and sparse vegetation region resulted by human and agricultural development. Figs. 4c, 4d, 4e and 4f demonstrate the two landform regions with different vegetation coverage. The landform region with sparse vegetation corresponds and is almost equal to the region of landslide. The UAS DSM generated in this study is very similar to so-called DEM, because the terrain is not concealed by the forest canopy. Thus, the geomorphologic analysis outside the landslide region depends mainly on the airborne LiDAR DSM and DEM in our study. Overall, the UAS and airborne LiDAR datasets can be mutually compensated for the geomorphological analysis in this study."

Referee #1-3. The volume of the CSL was six times larger than that of the largest landslide ever reported in Taiwan which was triggered by the Chi-Chi earthquake. How did the authors assume that the CSL triggered by the single earthquake event? Additionally, the authors assumed that current topography in the CSL corresponded to the slip surface of the original landslide (Fig. 13). Did authors have geological evidences of that? Detection of the slip surface is important for estimating the volume.

Response #1-3: Thank you for the helpful comments. To avoid misleading the readers, we have revised and added some texts in the manuscript. We don't think that the entire cut-and-fill volume of CSL was triggered only by one single event. However, from the geomorphologic features denoted in Fig. 10 (Zones A, B and C), the regions show different degrees of preservation of the landslide geomorphologic components. These observations suggest that more than one sliding event has occurred in the study area. Thus, the CSL can be interpreted to have occurred by a combination of multiple landslide events. Yet, it is difficult to propose how many landslide events have occurred in the study area. Regarding the landslide volume, the position and morphology of the slip surface indeed will affect the calculated cut-and-fill volume. In this study the slip surface is difficult to observe in the field due to soil cover and has not been definitely identified. Nevertheless, the sedimentary rock basement and the volcanic rock cover have been well mapped both on the geologic map and in field survey in the region (Fig. 1). Based on the distribution of rock types, it is supposed that the contact between the volcanic cover and the underneath sedimentary rocks may serve as a weak plane for the slip surface. On the other hand, the calculated landslide volume is derived from the difference of DEM, which denotes only the minimum volume, and does not take into account the remaining debris still resting on the supposed slip surface, especially for the larger landslide CSL. All the above-mentioned points are now improved in the text to avoid misleading information. The revised texts and in which sections of the manuscript are listed as follows:

– 4.2 Estimation of the landslide dimensions: "... The cut-and-fill volume is based on the difference of DEM, which indicates the minimum volume and does not account for the remaining debris on the slip surface. On the other hand, the volume does not consider how many landslide events have occurred to induce such volume due to insufficient evidence. " – 5 Discussion: "... However, from the geomorphologic features denoted in Fig. 10 (Zones A, B and C), the regions show different degrees of preservation of the landslide geomorphologic components. The CSL can be interpreted to have occurred from a combination of multiple landslide events. In

addition, the CSL and XSL preserved different degrees of landslide geomorphologic components, and the creeks as illustrated in Figs. 5 and 11 developed within the depletion zone with different drainage patterns and varying incision depths. These observations suggest that more than one sliding event has occurred in the study area. . . . . . . The remaining displaced material in the CSL suggests a combination of multiple landslide events. However, most of the displaced material in the XSL has been eroded away and it is not possible to estimate how many events are involved in the XSL. . . . Regarding the landslide volume, the position and morphology of the slip surface indeed will affect the calculated cut-and-fill volume. In this study the slip surface is difficult to observe in the field due to soil cover and has not been definitely identified. Nevertheless, the sedimentary rock basement and the volcanic rock cover have been well mapped both on the geologic map and in field survey in the region (Fig. 1). Based on the distribution of rock types, it is supposed that the contact between the volcanic cover and the underneath sedimentary rocks may serve as a weak plane for the slip surface. On the other hand, the calculated landslide volume is derived from the difference of DEM, which denotes only the minimum volume, and does not take into account the remaining debris still resting on the supposed slip surface, especially for the larger landslide CSL, as shown in Fig. 10."

Please also note the supplement to this comment:
https://www.nat-hazards-earth-syst-sci-discuss.net/nhess-2017-227/nhess-2017-227-AC1-supplement.pdf

---

## Referee Comment (RC2) · Anonymous Referee #2 · 11 Sep 2017

The study by Chang et al. investigates two large landslides developed along an active normal fault in a volcanic environment. Starting from previous knowledge about two large landslides in the area, the authors build their study on mapping the two landslides from visual interpretation of UAS imagery, as well as the interpretation of high-resolution digital topography (1 x 1 m LiDAR DEM). Based on their mapping, they estimate the volume of the two landslides by subtracting the present day topography from a reconstructed pre-failure topography. They conclude that the volume obtained is six times higher than the reported largest landslide volume in Taiwan. They further

postulate that an active normal fault controlled the morphological evolution of the two landslides, and that ongoing faulting is responsible for maintaining landslide hazard condition in the study area. While it is interesting the attempt of the authors to relate landslide evolution directly to fault activity, I'm not fully convinced by the story they want to tell. I identified many issues and problems with the data (1), methods (2), and interpretations (3) that preclude this from being a convincing study. These include lack of clarity in data and methods and what was actually measured, issues with the interpretations and what the data mean, and a lack of depth in the interpretations and implications that are drawn from the data.

1) I have reservations about some of the assumptions that the authors have gone into their dataset. In particular, I don't know where their slip surfaces position estimates have come from. These are critical, because it is the postulated spatial coincidence between the slip surfaces and the present-day topography that provides the condition to calculate the landslides volume according to the method presented in the paper. The authors are not clear at this point: only short and general shrift are done at lines 15-20 page 8, but without any geological evidence or examples, it's hard to know what, exactly, they have considered for their assumption. Geology of the area is presented in figure 1, but the figure is not informative enough to support the assumption of the authors. Clearly, the present day topography is somehow related to the movement along the slip surfaces, but I think the authors need to be a lot more careful about what they say, and do a better job of documenting why the present day topography can be considered the slip surface of an old landslide. I also have reservation about the landslide detection, mapping and classification. Figure 5 illustrate the detection of zones affected by mass movements highlighted by ridges and scarps, which are commonly interpreted as the topographic response to movements along the slip surfaces at depth. However, the evidences strongly contrast with the assumption done by the authors about the coincidence between the slip surface and the present-day topography. This is a main issue that the authors should address to be their contribution convincing. In addition, I have reservations about the mapping itself. Landslide mapping should include the definition of the scarp area, deposit area, and both the flanks (see for instance Santangelo et al. 2015 NHESS, 15, 2111–2126; Guzzetti et al. 2012, Earth Science Reviews, 112, 42-66; Ambrosi and Crosta, 2006, Engineering Geology, 83, 183-200). Looking Figure 5, I really don't know where the limits (even supposed) of the two landslides are positioned. The circumstance undermine the possibility to visually appreciate and to quantitatively measure landslide area in map. Furthermore, the paper is not informative enough about the landslide type, landslide age (even relative age) and different generation of landslides recognized inside the old landslides. The information is necessary to characterize the landslide morphology, evolution and hazard, which are specific purposes of the paper. I think a more detailed mapping using the high quality materials (UAS imagery and LiDAR DEM) available to the authors should be add to the paper.

2) Although the method seems to be reasonable in theory, too many issues remain unexplained. For instance: I disagree with the assumption that detailed UAV imagery are better than aerial photographs and/or satellite images to detect and characterized large landslides. My own experience suggest quite the opposite. Indeed, UAV imagery and detailed LiDAR DEM are very useful to perform detailed studies. As a matter of fact, one of the more interesting piece of work in the paper is related to the characterization of the micro-topography of the landslides and the discussion about the possibility to apply the method to the study of gully erosion. However, gully erosion appear to be as a minor complication compared to the estimation of the landslide volume of a giant landslide. Complication is irrelevant here if the authors focus their paper on the calculation of the total landslide volume.

3) The final interpretation is not convincing and rise many question: Why just such two landslides developed along a regional normal fault? What about other places along the fault? There is somethings peculiar in the specific location of the two landslides? (i.e. relative relief higher respect to other places along the fault?) geo-structural setting different respect to other places along the fault and prone to landslides? cluster of

strong earthquakes? evidence of high vertical deformation rates? what else?) In the scheme proposed by the authors the fault is the main factor controlling both the onset and the disruption of the landslides, but no analysis support their conclusion. I have also reservation about the idea that normal fault activity has the effect of cancel the landslide signature (third diagram in the final scheme). I think quite the opposite; fault activity sustain relief formation, maintaining the condition for landslide development (see Bucci et al. 2016, ESPL, 41, 711-720; and Densmore et al. 1997, Science 275, 369-72). The authors conclude somethings similar at lines 27-29 page 12, but their statement conflict with the idea illustrated in the scheme. Finally, the authors never explicitly address time scales of the considered landslides and fault, as well as the probable mismatch in timescale of the landsliding and faulting processes.

Finally, I have reservation about the general organization of the paper.

The chapter Introduction is a blend (sometime confused) of general issues about landslide identification and characterization. I suggest to restructure the text, developing a sharper motivation with some clearer objectives. Also, quote the pertinent literature addressing the mapping and analysis of large landslides. Pertinent local literature help understanding the state of the art at local scale. The authors are not clear enough at this point. For instance at line 25 page 2 the authors acknowledge that the two landslides were already recognized. So why the authors define the two landslides as "obscure" if they were already recognized? I think additional information should be provided, and a comparison of previous and new results should be done. Similarly, the manuscript lacks of references to international literature addressing mapping and analysis of large landslide in active regions. Pertinent international literature help defining the framework of the study and it should be quoted along the paper (see for instance Bucci et al. 2016, ESPL, 41, 711-720; Scheingross et al. 2013, Geological Society of America Bulletin, 125, 473-489; Bucci et al. 2013, Physics and Chemistry of the Earth, 63, 12–24; Strecker M.R. and Marret R. 1999, Geology, 27, 307-310)

The chapter geological background (lines 14-23 page 3) is confused: it is hard to follow

and to understand the polygenic history of the faults of the area. The chapter contain information negligible for the aim of the paper. At the same time, the chapter lack of potentially useful information about the age and deformation rate of active structures, seismicity, landslide events. Finally, lines 3-11 page 4 belong to method, not to geological background.

The chapters 3 and 4 mix up methods, results and discussion, which is also included in the following chapter: Discussion. This writing setting makes reading hard to follow and to understand. Please change the text of the manuscript including the following chapters: Methods (include here technical issues regarding UAS imagery, digital topography (1 x 1 m LiDAR DEM), how you define landslides, what do you map using conventional approach (i.e. stereoscopic aerial photo-interpretation), what new using UAS imagery and LiDAR DEM (would be good to see in map the differences), how you estimate the landslide dimension, how you carried out the morphological reconstruction); Results (includes the new data and maps); and then Discussion (what can we learn from the new data and what is the meaning also comparing to other works) and Conclusions (take home messages in short).

The chapters Discussion and Conclusion focus on the evolution of the two landslides, stressing the role of tectonics. However, the paper do not contain any new information/analysis/result related to tectonics. The evolution scheme drawn by the authors remain poorly constrained also by the lacks of geological evidences supporting the supposed coincidence of the slip surfaces and the present day topography. I suggest to reconsider in depth (or to drop) the part of the analysis related to the volume calculation of the two landslide, because it simply raises too many questions.

Apart the many issues and problems, the figures are good and the geomorphic application related to gully incision and related erosion of old landslides seems interesting, and I would like to eventually see it in print. I think the authors need to be more careful about what they claim, and more explicit about how they explain and relate their various data sets. If the authors could do a better job of documenting it, then the contribution

could be considered for publication after careful major revision.

---

## Author Comment (AC2) · 10 Oct 2017

Thank you for the helpful comments on our manuscript. Please find below our response and modifications that we have revised in the manuscript following the referee's comments and suggestions.

Anonymous Referee #2 The study by Chang et al. investigates two large landslides developed along an active normal fault in a volcanic environment. Starting from previous knowledge about two large landslides in the area, the authors build their study on

mapping the two landslides from visual interpretation of UAS imagery, as well as the interpretation of high-resolution digital topography (1 x 1 m LiDAR DEM). Based on their mapping, they estimate the volume of the two landslides by subtracting the present day topography from a reconstructed pre-failure topography. They conclude that the volume obtained is six times higher than the reported largest landslide volume in Taiwan. They further postulate that an active normal fault controlled the morphological evolution of the two landslides, and that ongoing faulting is responsible for maintaining landslide hazard condition in the study area. While it is interesting the attempt of the authors to relate landslide evolution directly to fault activity, I'm not fully convinced by the story they want to tell. I identified many issues and problems with the data (1), methods (2), and interpretations (3) that preclude this from being a convincing study. These include lack of clarity in data and methods and what was actually measured, issues with the interpretations and what the data mean, and a lack of depth in the interpretations and implications that are drawn from the data.

Referee #2-1. I have reservations about some of the assumptions that the authors have gone into their dataset. In particular, I don't know where their slip surfaces position estimates have come from. These are critical, because it is the postulated spatial coincidence between the slip surfaces and the present-day topography that provides the condition to calculate the landslides volume according to the method presented in the paper. The authors are not clear at this point: only short and general shrift are done at lines 15-20 page 8, but without any geological evidence or examples, it's hard to know what, exactly, they have considered for their assumption. Geology of the area is presented in figure 1, but the figure is not informative enough to support the assumption of the authors. Clearly, the present day topography is somehow related to the movement along the slip surfaces, but I think the authors need to be a lot more careful about what they say, and do a better job of documenting why the present day topography can be considered the slip surface of an old landslide. I also have reservation about the landslide detection, mapping and classification. Figure 5 illustrate the detection of zones affected by mass movements highlighted by ridges and scarps, which

are commonly interpreted as the topographic response to movements along the slip surfaces at depth. However, the evidences strongly contrast with the assumption done by the authors about the coincidence between the slip surface and the present-day topography. This is a main issue that the authors should address to be their contribution convincing. In addition, I have reservations about the mapping itself. Landslide mapping should include the definition of the scarp area, deposit area, and both the flanks (see for instance Santangelo et al. 2015 NHESS, 15, 2111–2126; Guzzetti et al. 2012, Earth Science Reviews, 112, 42-66; Ambrosi and Crosta, 2006, Engineering Geology, 83, 183-200). Looking Figure 5, I really don't know where the limits (even supposed) of the two landslides are positioned. The circumstance undermine the possibility to visually appreciate and to quantitatively measure landslide area in map. Furthermore, the paper is not informative enough about the landslide type, landslide age (even relative age) and different generation of landslides recognized inside the old landslides. The information is necessary to characterize the landslide morphology, evolution and hazard, which are specific purposes of the paper. I think a more detailed mapping using the high quality materials (UAS imagery and LiDAR DEM) available to the authors should be add to the paper.

Response #2-1: We have reconstructed the above section into 5 separate questions (a to e), and responded these questions accordingly:. a) The slip surfaces and the present-day topography that provides the condition to calculate the landslides volume according to the method presented in the paper, but without any geological evidence or examples. Response #2-1a: Indeed, to estimate the landslide volume, the original topography and slip surface are the key issues. However, with regarding to an old landslide, the original surface is unknown. On the other hand, slip surface is usually covered by the slid mass, and is not easily exposed. Therefore, in this study we try to propose one of the methods to reasonably construct reasonably by the original ground surface and assume the location of the slip surface that likely located at the interface between the volcanic cover and the underlying sedimentary rocks. The original ground surface is constructed from ideal volcano cone edifice. The sedimentary rock basement

and the volcanic rock cover have been well mapped both on the geologic maps (Fig. 1 and the new Fig. 5) and in field survey in the region. Based on the distribution of rock types, it is supposed that the contact between the volcanic cover and the underneath sedimentary rocks may serve as a weak plane for the slip surface. The slip surface consists from the difference of material and the exposed different lithology.

We have revised and newly improved the paragraph in the manuscript. b) Geology of the area is not informative enough to support the assumption of the authors, and do a better job of documenting why the present day topography can be considered the slip surface of an old landslide. Response #2-1b: We have now added a figure to show more detailed local geologic conditions. In the new geological map, many landslides that occurred in the study area and in the Tatun Volcano region were attached to demonstrate the distribution of landslides. Compare Comparing the size, distribution and classification, etc. the two largest landslides (XSL and CSL) were thus chosen as the target for this study. We have newly revised and improved the paragraph in the manuscript. On the other hand, the 2D hillshade map (the original Fig. 5, now change to Fig. 6) has been modified with the azimuth of shade illumination to being 315° so as to better illustrate the landslide geomorphologic features.

c) Figure 5 illustrate the detection of zones affected by mass movements highlighted by ridges and scarps, which are commonly interpreted as the topographic response to movements along the slip surfaces at depth. However, the evidences strongly contrast with the assumption done by the authors about the coincidence between the slip surface and the present-day topography. This is a main issue that the authors should address to be their contribution convincing. Response #2-1c: The original Fig. 5 is now modified as Fig. 6. Indeed, ridges and scarps of a landslide are commonly interpreted as the topographic response of the movements along the slip surfaces at depth. However, the topographic feature responses reflect only the ground subsidence actually. Thus if the slid mass glides with a long run out distance or the displaced mass has been eroded away, both processes will preserved topographic remain relics by

distinct shutter ridges and scarps. In consequence, in this study we interpret that most of the material has been eroded away, and discussed from the perspectives of normal faulting and tectonic setting of the study area. We have newly improved the manuscript to better illustrate the full overall framework of the study.

d) Looking Figure 5, I really don't know where the limits (even supposed) of the two landslides are positioned. The circumstance undermine the possibility to visually appreciate and to quantitatively measure landslide area in map. Response #2-1d: The original figure Fig. 5 (now modified as Fig. 6) is has been modified with the azimuth of shade illumination to being 315° so as to better illustrate the landslide geomorphologic features. This new hillshade image should shall improve essentially the identification of landslide region visually, since because not necessary that all readers are familiar with the landslide morphology.

e) The paper is not informative enough about the landslide type, landslide age (even relative age) and different generation of landslides recognized inside the old landslides. Response #2-1e: The normal faulting in the region started since from 400Ka, and is activated continuously ever since. The faulting being was identified in the Taipei basin area and northeastern offshore Taiwan, with the fault line situated on both two sides of the study area. And the fault line was recently identified and linked together as only one normal fault in Tatun Volcano region (near and surrounding the study area) by the authors. In conclusion, for the relative age of the landslide, we interpret that the landslide has been triggered since right after normal faulting started and the formation of Tatun Volcano, say which is far later than 200 Ka. Regarding to the different generation of landslide, the geomorphologic components show different degrees of preservation within the two observed landslides. Furthermore, the CSL is interpreted to have occurred from a combination of multiple landslide events. We have newly improved revised the manuscript to denote the relative age of the landslide, and the different generation of landslidesas well.

Referee #2-2. Although the method seems to be reasonable in theory, too many is-

sues remain unexplained. For instance: I disagree with the assumption that detailed UAV imagery are better than aerial photographs and/or satellite images to detect and characterized large landslides. My own experience suggest quite the opposite. Indeed, UAV imagery and detailed LiDAR DEM are very useful to perform detailed studies. As a matter of fact, one of the more interesting piece of work in the paper is related to the characterization of the micro-topography of the landslides and the discussion about the possibility to apply the method to the study of gully erosion. However, gully erosion appear to be as a minor complication compared to the estimation of the landslide volume of a giant landslide. Complication is irrelevant here if the authors focus their paper on the calculation of the total landslide volume.

Response #2-2: We have reconstructed divided the above section into 2 questions (a to b), and response responded the questions accordingly: a) I disagree with the assumption that detailed UAV imagery are better than aerial photographs and/or satellite images to detect and characterized large landslides. Response #2-2a: In Taiwan, on the one hand heavy precipitation induced by the annual northeast monsoon markedly modifies easily the landslide topography. On the other hand, the study region is situated within a national park and preserves well dense forest very well. Both effects conceal detaiedl topography and nearly impossible to study directly from aerial photographs and/or satellite images. The same situation can be found in the Tsaoling and Jiufengershan two giant landslides (namely, Tsaoling and Jiufengershan) triggered by the Chi-Chi earthquakes, where the vegetation colonization cover and concealed almost all the topographic feathersdetails, especially for the zone of accumulationn, in just only ten years after the landslides has been occurred. That is why we deploy employedtwo high- resolution and high- precision date datasets/methods, the UAV and the airborne LiDAR, to decipher the landslide features of the study area. And that is why we insistassert the quality and different levels of the data sets, and illustrated and denotethemd in Ffigs. 2 and 4. We have newly improved revised and clarified the documentation in the manuscript.

b) Gully erosion appears to be as a minor complication compared to the estimation of the landslide volume of a giant landslide. Complication is irrelevant here if the authors focus their paper on the calculation of the total landslide volume. Response #2-2b: Yes, the gully incision is a minor factor to estimate the overall landslide volume. The method is used only to assess the landslide morphology and evolution. We have clarified the documentation in the manuscript.

Referee #2-3. The final interpretation is not convincing and rise many question: Why just such two landslides developed along a regional normal fault? What about other places along the fault? There is somethings peculiar in the specific location of the two landslides? (i.e. relative relief higher respect to other places along the fault?) geo-structural setting different respect to other places along the fault and prone to land-slides? cluster of strong earthquakes? evidence of high vertical deformation rates? what else?) In the scheme proposed by the authors the fault is the main factor control-ling both the onset and the disruption of the landslides, but no analysis support their conclusion. I have also reservation about the idea that normal fault activity has the effect of cancel the landslide signature (third diagram in the final scheme). I think quite the opposite; fault activity sustain relief formation, maintaining the condition for land-slide development (see Bucci et al. 2016, ESPL, 41, 711-720; and Densmore et al. 1997, Science 275, 369-72). The authors conclude somethings similar at lines 27-29 page 12, but their statement conflict with the idea illustrated in the scheme. Finally, the authors never explicitly address time scales of the considered landslides and fault, as well as the probable mismatch in timescale of the landsliding and faulting processes.

Response #2-3: We have reconstructed divided the paragraph by into 3 questions (a to c) and responded se accordingly: a) Why just such two landslides developed along a regional normal fault? What about other places along the fault? There is somethings peculiar in the specific location of the two landslides? (i.e. relative relief higher respect to other places along the fault?) geo-structural setting different respect to other places along the fault and prone to landslides? cluster of strong earthquakes? evidence of

high vertical deformation rates? what else?) Response #2-3a: In northern Taiwan, the tectonic activity of the region is in extensional regionregime, thus dominated by normal faulting in the study area nowadays. The Jinshan fault (JSF), and Shanchiao fault (SCF, also known as the Jinshan Fault through with normal faulting mechanism), both of the faulting was were being identified longtime ago in Taipei Basin area (Southwest to the study area) and in northeastern offshore Taiwan (northeast of the study area). And recently these two faults were identified and to have linked together as only by only one normal fault in the Tatun Volcano region (around and across the study area ) by the authors. (tThe result was published in the Central Geological Survey CGS project report written in Chinese, and the paper for international journal is now in preparation). On the other hand, there are many landslides within the study area and in the Tatun Volcano region, as shown in the newly added figure Fig. 5. Compare Comparing the size, distribution and classification, etc. the two largest landslides (XSL and CSL) were thus chosen as the target for this study. We have newly revised and improved the paragraph in the manuscript.

b) I have also reservation about the idea that normal fault activity has the effect of cancel the landslide signature (third diagram in the final scheme). I think quite the opposite; fault activity sustain relief formation, maintaining the condition for landslide development. The authors conclude somethings similar at lines 27-29 page 12, but their statement conflict with the idea illustrated in the scheme. Response #2-3b: In northern Taiwan, the tectonic activity of the region is in extensional regionregime. The Jinshan normal faulting cause resulted in the formation of Taipei basin by over one thousand meter throw of the fault separation. The normal faulting is has been very well documented recently, e.g. Teng et al., (2001); Shyu et al., (2005); C.T. Chen et al., (2007, 2010); Huang et al., (2007); and K.C. Chen et al., (2010). And this normal faulting caused the formation of the Taipei basin and may also cause also the continuous eruption of the Tatun Volcano. The evidence of normal faulting being has been recently identified in Taipei basin area and northeastern offshore Taiwan. And recently identified and linked together as only one normal fault in Tatun Volcano region

(near and around the study area) by the authors. And two original normal faults are considered to be linked together as a long stretched normal fault that may provide significant earthquake faulting. Finally, the total length of the Jinshan normal fault is over more than as long as 130 Km long. In conclusions, weWe thus interpret that the normal faulting leads has led to the formation of the slope daylight, as well as the volcano subsidence in southern the south of the study area. This process may likely leads to the formation of the landslide. As Because the normal faulting activated continuously, the sliding mass may be transporting continuously to the Jinshan Delta. The fig original Fig. 13 (now modified as Fig. 14) demonstrates ideally (of course not fully adapted the current topography) the general geomorphologic evolution ideally, and so as to explain the wear off of the landslide deposits, especially in XSL.

c) The authors never explicitly address time scales of the considered landslides and fault, as well as the probable mismatch in timescale of the landsliding and faulting. Response #2-3c: The normal faulting started since from 400 Ka, and activated continuously ever since. The age of the Tatun volcano is smaller than 200 Ka. So the relative age of the landslide is been triggered sincemost probably after the normal faulting and the formation of the Tatun Volcano, say farwhich is later than 200 Ka. On the other hand, the CSL and XSL preserve different degrees of landslide geomorphologic components, showing the sitesa combination of multiple landslide events. Furthermore, part of the fault branches is identified on the lower slope within the sliding area, express prompting the faulting behavior truncates and enhances the erosion process. In conclusion, based on many aspects, thus the authors thus interpret propose one model to access highlight the possiblethe landslide evolution that will be useful for further testing.

Referee #2-4. Finally, I have reservation about the general organization of the paper. The chapter Introduction is a blend (sometime confused) of general issues about landslide identification and characterization. I suggest to restructure the text, developing a sharper motivation with some clearer objectives. Also, quote the pertinent literature

addressing the mapping and analysis of large landslides. Pertinent local literature help understanding the state of the art at local scale. The authors are not clear enough at this point. For instance at line 25 page 2 the authors acknowledge that the two landslides were already recognized. So why the authors define the two landslides as "obscure" if they were already recognized? I think additional information should be provided, and a comparison of previous and new results should be done. Similarly, the manuscript lacks of references to international literature addressing mapping and analysis of large landslide in active regions. Pertinent international literature help defining the framework of the study and it should be quoted along the paper (see for instance Bucci et al. 2016, ESPL, 41, 711-720; Scheingross et al. 2013, Geological Society of America Bulletin, 125, 473-489; Bucci et al. 2013, Physics and Chemistry of the Earth, 63, 12–24; Strecker M.R. and Marret R. 1999, Geology, 27, 307-310) The chapter geological background (lines 14-23 page 3) is confused: it is hard to follow and to understand the polygenic history of the faults of the area. The chapter contain information negligible for the aim of the paper. At the same time, the chapter lack of potentially useful information about the age and deformation rate of active structures, seismicity, landslide events. Finally, lines 3-11 page 4 belong to method, not to geological background. The chapters 3 and 4 mix up methods, results and discussion, which is also included in the following chapter: Discussion. This writing setting makes reading hard to follow and to understand. Please change the text of the manuscript including the following chapters: Methods (include here technical issues regarding UAS imagery, digital topography (1 x 1 m LiDAR DEM), how you define landslides, what do you map using conventional approach (i.e. stereoscopic aerial photo-interpretation), what new using UAS imagery and LiDAR DEM (would be good to see in map the differences), how you estimate the landslide dimension, how you carried out the morphological reconstruction); Results (includes the new data and maps); and then Discussion (what can we learn from the new data and what is the meaning also comparing to other works) and Conclusions (take home messages in short). The chapters Discussion and Conclusion focus on the evolution of the two landslides, stressing the role of tectonics. However,

the paper do not contain any new information/analysis/result related to tectonics. The evolution scheme drawn by the authors remain poorly constrained also by the lacks of geological evidences supporting the supposed coincidence of the slip surfaces and the present day topography. I suggest to reconsider in depth (or to drop) the part of the analysis related to the volume calculation of the two landslide, because it simply raises too many questions.

Response #2-4: We have reconstructed divided the above paragraph by into 4 questions (a to d), and response responded accordingly:

a) I suggest to restructure the text, developing a sharper motivation with some clearer objectives. Also, quote the pertinent literature addressing the mapping and analysis of large landslides. Pertinent local literature help understanding the state of the art at local scale. The authors are not clear enough at this point. For instance at line 25 page 2 the authors acknowledge that the two landslides were already recognized. So why the authors define the two landslides as "obscure" if they were already recognized? I think additional information should be provided, and a comparison of previous and new results should be done. Similarly, the manuscript lacks of references to international literature addressing mapping and analysis of large landslide in active regions. Pertinent international literature help defining the framework of the study and it should be quoted along the paper. Response #2-4a: Some of the p Pertinent literatures were are now added into the manuscript. The two landslides were already recognized from geomorphologic 40 m DTM by Prof. C. T. Lee of the National Central University (from only personal communication). However, due to the lake lack of existed/available datasets and no districtwithout distinct features, the landslides was were not been analyzed deeply in depth till this study. From climatologic point of view, the annual rainfall is more than 2500 mm in this area, thus a vast portion of the study area is covered by vegetation. Dense forest thus partially conceals morphological features and has prevented detailed geomorphic studies in the past. On the other hand, the heavy rainfall also increases enhances the surface processes, e.g., incision and erosion. As a consequence, the erosion effect also obscures the landslide features in addition. To all of the points, weWe have newly improved the documentation in the manuscript based on the abovementioned points.

b) The chapter geological background (lines 14-23 page 3) is confused: it is hard to follow and to understand the polygenic history of the faults of the area. The chapter contain information negligible for the aim of the paper. At the same time, the chapter lack of potentially useful information about the age and deformation rate of active structures, seismicity, landslide events. Response #2-4b: To discuss the landslide evolution, especially for an old landslide, the geologic and regional tectonics must been be included, even for the factor of surface process, e.g. climate etc. So theThe polygenic history of the study area must been be taken into account. In the study area, so we consider many factors, including, : lithology, normal fault, climate, vegetation, erosion and human agriculture activity etc., so asin order to access the landslide geomorphologic evolution. Regarding to the slip rate of Jinshan normal faulting, it is shown between 8.2-1.8 mm/yr subsiding rate in at different sites and in time intervals (, e.g., Rau et al., 2006; Huang et al., 2007; Chen et al., 2010). This high slip rate creates the Taipei Basin, and may significantly affects importantly the landslide evolution as well. But unfortunately, these slip rate studiesy is were focused only on the Taipei Basin, and not on the study area. The manuscript is now newly reinforced and improved to clarify the tectonic factor and the interaction.

c) This writing setting makes reading hard to follow and to understand. Please change the text of the manuscript including the following chapters: Methods; Results; and then Discussion and Conclusions. Response #2-4c: We have newly improved the manuscript according to the comment.

d) The chapters Discussion and Conclusion focus on the evolution of the two landslides, stressing the role of tectonics. However, the paper do not contain any new information/analysis/result related to tectonics. The evolution scheme drawn by the authors remain poorly constrained also by the lacks of geological evidences support-

ing the supposed coincidence of the slip surfaces and the present day topography. Response #2-4d: One geological map (Fig. 5) have has been added to demonstrate the geological background of study area, and to better linked the relationship between the regional tectonics and landslide geology and evolution. The manuscript has been improved accordingly.

Please also note the supplement to this comment:
https://www.nat-hazards-earth-syst-sci-discuss.net/nhess-2017-227/nhess-2017-227-AC2-supplement.pdf

**Supplement:**

[revised manuscript text omitted]

---

## Author Response (AR1)

We thank very much the two referees' and the editor's helpful and constructive comments on our manuscript. Please find below our responses and modifications that we have revised in the manuscript according to the reviews. For completeness, we have included all the referees' and the editor's comments and responded the comments accordingly on a point-by-point basis as follows.

**Comments by Editor Prof. Hayakawa**

Dear Authors,

Your manuscript "Geomorphological evolution of landslides near an active normal fault in Northern Taiwan, as revealed by LiDAR and unmanned aircraft system data " (nhess-2017-227) has been assessed by two reviewers. They raised several critical issues that need to be fixed before publication, and you have provided detailed feedbacks during the NHESS open discussion. Based on this discussion, and my own assessment as Editor, I would like to inform you that it is potentially acceptable for publication in NHESS, provided that you carry out essential revisions suggested by the reviewers.

The paper is interesting, but it falls on the borderline regarding the scope of NHESS. The work focuses more on geomorphology with the long-term development of landforms while less related to hazards. Although the authors discuss some of the activities of the normal fault in recent years, the importance of this work is unclear for the formation and development of the gigantic landslides with the fault activities. I think discussions in this line can be further developed.

Once you have made the necessary corrections, please submit a revised manuscript with point-by-point responses to both reviewers, and highlight all changes made when revising the manuscript. Please ensure you include a detailed rebuttal of any criticisms or requested revisions that you disagreed with. Also, ensure that your revised manuscript conforms to the journal style.

Please note that this editorial decision does not guarantee that your manuscript will be accepted for final publication in NHESS. A decision will be made after the revised version is submitted and evaluated by the same or further reviewers.

We look forward to receiving your revised manuscript soon.

Best regards,

yuichi hayakawa

**End of Comments by Editor Prof. Hayakawa**

<Our Response to the Editor's Comments>: Thank you very much for the overall

comments and suggestions to our manuscript. We have now modified and improved our manuscript according the comments by the two reviewers and strengthened the view of natural hazards by large landslides that might be caused by active fault activities and indicated by the geomorphological evolution as revealed by the landforms. As the journal "Natural Hazards and Earth Sciences System (NHESS)" suggested, we would also like to emphasize the relationship between the large landslides and the interaction between the Earth tectonic system and geomorphological system in the long term. Because landslide hazard study is a multi-disciplinary science, it is proper to include many aspects, including geomorphological features and evolution, so as to decipher the complex behavior that may eventually lead to the failure of the landslide. In this study, we relied on the analysis of landslides and tried to link the results with the regional tectonics/faulting activities that may trigger the landslides that could cause a serious hazard if similar situation happens in the future. In the work, we provide not only the landslide hazard but also the full scope of the mechanism and the evolution of landslides. We have emphasized more clearly on the point in the renewed manuscript. The UAS techniques used in this study covered a relatively large area. We then compared and verified with different sources, including Airborne LiDAR, to include not only the hazard application but also validation of the UAS-derived imagery as it will be useful for future UAS application in natural hazards. The importance of this work lies in the formation and development of the landslides potentially accompanied with the active fault activities. We have revised and improved the manuscript, so that it is now better fitted to the scopes as mentioned. All the modifications are highlighted in a different color in the newly revised manuscript, and all the responses for the two reviewers are listed point-by-point in the following in the attached file. We appreciate very much again for your constructive input at improving the current manuscript.

**< End of Our Response to the Editor's Comments>**

**Comments by Anonymous Referee #1**

This study analyzed landslide morphological characteristics and geomorphological evolution using lidar and UAS data in northern Taiwan. The morphological reconstruction showed that the total volume of landslides reached 820 x 10ˆ6 mˆ3. This paper is interesting for the evaluation of landslide evolution and the assessment of related landslide hazards. However, the reviewer has some comments regarding landslide types, data, and methods that would need to be verified by authors.

**#1-1.** Little information on landslide types in the study area was explained in the manuscript. Landslide types are important for discussing the landslide evolution. It would be better to show landslide types and processes analyzed in this study referring Varnes (1978) or Cruden and Varnes (1996).

**<Our Response #1-1>:** According to the criteria of landslide classification proposed by Varnes (1978), the two major landslides analyzed in this study may be classified as rotational or translational slide. Detailed characteristics of the two slide types commonly include circular crown, main scarp, minor scarps, circular transverse ridges and lateral franks. We have newly added a paragraph in the manuscript to explain the observed landslide types in the study area. On the other hand, the 2D hillshade map (the original Fig. 5, now Fig. 6) has been modified with the azimuth of shade illumination being 315° to better illustrate the landslide geomorphologic features. The paragraph is as follows: "As mentioned and illustrated in Figs. 6 and 7, the CSL is marked with circular crown, main scarp, circular concentric transverse ridges in the rear of the main body, whereas, most of the landslide morphologic components in the XSL have been modified by human activities. For example, the crown area of the landslide has been developed into a graveyard with clearly preserved lateral franks. According to the criteria of landslide classification proposed by Varnes (1978), the two major landslides, from the currently observed landslide geomorphologic components, suggest that the landslides are best classified as rotational or translation slides."
 **<End of Our Response #1-1>**

**#1-2.** The authors emphasized importance of UAS and lidar data. However, it was not clear how did authors use these DSMs for the geomorphological analysis, respectively. For example, the authors explained that USA had the disadvantage that the DSMs included the vegetation height. How did authors use the DSMs for the analysis? Were the geomorphological analysis and the reconstruction performed by lidar data alone?

**<Our Response #1-2>:** Both the airborne LiDAR and UAS datasets are used in this study. The UAS DSM and the orthomosaic photos generated in this study are as high as 8.5 cm in pixel resolution, thus, the ground information is much more easily identified for regions of building and sparse vegetation. Especially, within the two major landslides CSL and XSL, the terrains have been affected by human and

agricultural development, which is marked by a terrain with minor and low vegetation. The UAS DSM dataset is useful for processing and separating the DSM and DEM derived from airborne LiDAR dataset, because the UAS DSM is most informative at distinguishing the ground facts. In response to the comment, the above-mentioned points are added in the revised manuscript as follows: "The UAS images, which generate 8.5 cm pixel resolution in both the orthomosaic photo and DSM, distinguish clearly the ground and non-ground features, such as buildings and sparse vegetation. Moreover, this information is helpful at improving the airborne LiDAR data processing and point clouds classification. In the study area, two different landforms can be readily distinguished, i.e., dense forest and sparse vegetation region resulted by human and agricultural development. Figs. 4c, 4d, 4e and 4f demonstrate the two landform regions with different vegetation coverage. The landform region with sparse vegetation corresponds and is almost equal to the region of landslide. The UAS DSM generated in this study is very similar to so-called DEM, because the terrain is not concealed by the forest canopy. Thus, the geomorphologic analysis outside the landslide region depends mainly on the airborne LiDAR DSM and DEM in our study. Overall, the UAS and airborne LiDAR datasets can be mutually compensated for the geomorphological analysis in this study."

  **<End of Our Response #1-2>**

**#1-3.** The volume of the CSL was six times larger than that of the largest landslide ever reported in Taiwan which was triggered by the Chi-Chi earthquake. How did the authors assume that the CSL triggered by the single earthquake event? Additionally, the authors assumed that current topography in the CSL corresponded to the slip surface of the original landslide (Fig. 13). Did authors have geological evidences of that? Detection of the slip surface is important for estimating the volume.

**<Our Response #1-3>:** Thank you for the helpful comments. To avoid misleading the readers, we have revised and added some texts in the manuscript. We don't think that the entire cut-and-fill volume of CSL was triggered only by one single event. Meanwhile, from the geomorphologic features denoted in Fig. 10 (Zones A, B and C), the regions show different degrees of preservation of the landslide geomorphologic components. These observations suggest that more than one sliding event has occurred in the study area. Thus, the CSL can be interpreted to have occurred by a combination of multiple landslide events. Yet, it is difficult to propose how many landslide events have occurred in the study area. Regarding the landslide volume, the

position and morphology of the slip surface indeed will affect the calculated cut-and-fill volume. In this study the slip surface is difficult to observe in the field due to soil cover and has not been definitely identified. Nevertheless, the sedimentary rock basement and the volcanic rock cover have been well mapped both on the geologic map and in field survey in the region (Fig. 1, and newly added Fig. 5). Based on the distribution of rock types, it is supposed that the contact between the volcanic cover and the underneath sedimentary rocks may serve as a weak plane for the slip surface. On the other hand, the calculated landslide volume is derived from the difference of DEM, which denotes only the minimum volume, and does not take into account the remaining debris still resting on the supposed slip surface, especially for the larger landslide CSL. All the above-mentioned points are now improved in the text to avoid misleading information. They are now added in the revised manuscript section 5.2, as follows:

5.2 Landslide slip surface and volume estimation:

According to geological data and field observations, the Tatun volcanic rocks overlie the Mio-Pliocene sedimentary rocks (Figs. 1 and 5). Because of the clear contrast in the rock strength and strata unconformity, the contact surface of the two rock types may easily serve as the rupture site for the sliding surface, thus indicating that most landslide debris were eroded or slid away when the contact surface was largely exposed. Thus, although the estimated volume was considered the minimum landslide volume, volume estimation shall approximate the actual volume in this special case.

Regarding the landslide volume, the position and morphology of the slip surface indeed will affect the calculated cut-and-fill volume. In this study the slip surface is difficult to observe in the field due to soil cover and has not been definitely identified. Nevertheless, the sedimentary rock basement and the volcanic rock cover have been well mapped both on the geologic map and in field survey in the region (Figs. 1 and 5). Based on the distribution of rock types, it is supposed that the contact between the volcanic cover and the underneath sedimentary rocks may serve as a weak plane for the slip surface. On the other hand, the calculated landslide volume is derived from the difference of DEM, which denotes only the minimum volume, and does not take into account the remaining debris still resting on the supposed slip surface, especially for the larger landslide CSL, as shown in Fig. 11.

For the XSL, the maximum cut depth was approximately 150 m. The maximum cut area was situated in the central zone of the sliding area and showed a symmetrical reverse-conic shape; a uniform erosion process in the accumulated area may account for this pattern. For the CSL, the sliding mass had a sliding depth of approximately

200 m and a wide and flat bottom. The volume of the CSL landslide was approximately three times larger than that of the XSL. Although the assumed ideal volcanic conical dome may deviate from the true shape of the topography, the estimated results provided useful information about ideal magnitudes of the scale and volumes of landslides, which are several times higher than the magnitudes previously reported in Taiwan.
**<End of Our Response #1-3>**

**End of Comments by Anonymous Referee #1**

**Comments by Anonymous Referee #2**

The study by Chang et al. investigates two large landslides developed along an active normal fault in a volcanic environment. Starting from previous knowledge about two large landslides in the area, the authors build their study on mapping the two landslides from visual interpretation of UAS imagery, as well as the interpretation of high-resolution digital topography (1 x 1 m LiDAR DEM). Based on their mapping, they estimate the volume of the two landslides by subtracting the present day topography from a reconstructed pre-failure topography. They conclude that the volume obtained is six times higher than the reported largest landslide volume in Taiwan. They further postulate that an active normal fault controlled the morphological evolution of the two landslides, and that ongoing faulting is responsible for maintaining landslide hazard condition in the study area. While it is interesting the attempt of the authors to relate landslide evolution directly to fault activity, I'm not fully convinced by the story they want to tell. I identified many issues and problems with the data (1), methods (2), and interpretations (3) that preclude this from being a convincing study. These include lack of clarity in data and methods and what was actually measured, issues with the interpretations and what the data mean, and a lack of depth in the interpretations and implications that are drawn from the data.

**#2-1.** I have reservations about some of the assumptions that the authors have gone into their dataset. In particular, I don't know where their slip surfaces position estimates have come from. These are critical, because it is the postulated spatial coincidence between the slip surfaces and the present-day topography that provides the condition to calculate the landslides volume according to the method presented in the paper. The authors are not clear at this point: only short and general shrift are done

at lines 15-20 page 8, but without any geological evidence or examples, it's hard to know what, exactly, they have considered for their assumption. Geology of the area is presented in figure 1, but the figure is not informative enough to support the assumption of the authors. Clearly, the present day topography is somehow related to the movement along the slip surfaces, but I think the authors need to be a lot more careful about what they say, and do a better job of documenting why the present day topography can be considered the slip surface of an old landslide. I also have reservation about the landslide detection, mapping and classification. Figure 5 illustrate the detection of zones affected by mass movements highlighted by ridges and scarps, which are commonly interpreted as the topographic response to movements along the slip surfaces at depth. However, the evidences strongly contrast with the assumption done by the authors about the coincidence between the slip surface and the present-day topography. This is a main issue that the authors should address to be their contribution convincing. In addition, I have reservations about the mapping itself. Landslide mapping should include the definition of the scarp area, deposit area, and both the flanks (see for instance Santangelo et al. 2015 NHESS, 15, 2111–2126; Guzzetti et al. 2012, Earth Science Reviews, 112, 42-66; Ambrosi and Crosta, 2006, Engineering Geology, 83, 183-200). Looking Figure 5, I really don't know where the limits (even supposed) of the two landslides are positioned. The circumstance undermine the possibility to visually appreciate and to quantitatively measure landslide area in map. Furthermore, the paper is not informative enough about the landslide type, landslide age (even relative age) and different generation of landslides recognized inside the old landslides. The information is necessary to characterize the landslide morphology, evolution and hazard, which are specific purposes of the paper. I think a more detailed mapping using the high quality materials (UAS imagery and LiDAR DEM) available to the authors should be add to the paper.

**\<Our Response #2-1\>:** We have divided the above section into 5 separate questions (a to e), and responded these questions accordingly.

a) "The slip surfaces and the present-day topography that provides the condition to calculate the landslides volume according to the method presented in the paper, but without any geological evidence or examples."

**Response #2-1a:** Indeed, to estimate the landslide volume, the original topography and slip surface are the key issues. However, regarding to an old landslide, the original surface is unknown. On the other hand, slip surface is usually covered by

the slid mass, and is not easily exposed. Therefore, in this study we try to propose one of the methods to reasonably construct the original ground surface and assume the slip surface that likely located at the interface between the volcanic cover and the underlying sedimentary rocks. The original ground surface is constructed from ideal volcano cone edifice. The sedimentary rock basement and the volcanic rock cover have been well mapped both on the geologic maps (Fig. 1 and the new Fig. 5) and in field survey in the region. Based on the distribution of rock types, it is supposed that the contact between the volcanic cover and the underneath sedimentary rocks may serve as a weak plane for the slip surface. The slip surface consists from the difference of material and the exposed different lithology. We have revised and improved the paragraph in the manuscript.

b) "Geology of the area is not informative enough to support the assumption of the authors, and do a better job of documenting why the present day topography can be considered the slip surface of an old landslide."

**Response #2-1b:** We have now added a figure to show more detailed local geologic conditions. In the new geological map, many landslides that occurred in the study area and in the Tatun Volcano region were attached to demonstrate the distribution of landslides. Comparing the size, distribution and classification, the two largest landslide (XSL and CSL) were thus chosen as the target for this study. We have revised and improved the paragraph in the manuscript. On the other hand, the 2D hillshade map (the original Fig. 5, now Fig. 6) has been modified with the azimuth of shade illumination being 315° to better illustrate the landslide geomorphologic features.

c) "Figure 5 illustrate the detection of zones affected by mass movements highlighted by ridges and scarps, which are commonly interpreted as the topographic response to movements along the slip surfaces at depth. However, the evidences strongly contrast with the assumption done by the authors about the coincidence between the slip surface and the present-day topography. This is a main issue that the authors should address to be their contribution convincing."

**Response #2-1c:** The original Fig. 5 is now modified as Fig. 6. Indeed, ridges and scarps of a landslide are commonly interpreted as the topographic response of the movements along the slip surfaces at depth. However, the topographic feature responses reflect only the ground subsidence actually. Thus if the slid mass glides with a long run out distance or the displaced mass has been eroded away, both

processes will preserved topographic relicts by distinct shutter ridges and scarps. In consequence, we interpret that most of the material has been eroded away from the perspectives of normal faulting and tectonic setting of the study area. We have newly improved the manuscript to better illustrate the overall framework of the study.

**d)** "Looking Figure 5, I really don't know where the limits (even supposed) of the two landslides are positioned. The circumstance undermine the possibility to visually appreciate and to quantitatively measure landslide area in map."

**Response #2-1d:** The original Fig. 5 (now Fig. 6) has been modified with the azimuth of shade illumination being 315° to better illustrate the landslide geomorphologic features. This new hillshade image shall improve the identification of landslide region visually, because not all readers are familiar with the landslide morphology.

**e)** "The paper is not informative enough about the landslide type, landslide age (even relative age) and different generation of landslides recognized inside the old landslides."

**Response #2-1e:** The normal faulting in the region started from 400Ka and is activated continuously ever since. The faulting was identified in the Taipei basin area and northeastern offshore Taiwan, with the fault line situated on both sides of the study area. And the fault line was recently identified and linked together as only one normal fault in Tatun Volcano region (near and surrounding the study area) by the authors. In conclusion, for the relative age of the landslide, we interpret that the landslide has been triggered since right after normal faulting started and the formation of Tatun Volcano, which is far later than 200 Ka. Regarding to the different generation of landslide, the geomorphologic components show different degrees of preservation within the two observed landslides. Furthermore, the CSL is interpreted to have occurred from a combination of multiple landslide events. We have newly revised the manuscript to denote the relative age of the landslide and the different generation of landslides.

**<End of Our Response #2-1>**

**#2-2.** Although the method seems to be reasonable in theory, too many issues remain

unexplained. For instance: I disagree with the assumption that detailed UAV imagery are better than aerial photographs and/or satellite images to detect and characterized large landslides. My own experience suggest quite the opposite. Indeed, UAV imagery and detailed LiDAR DEM are very useful to perform detailed studies. As a matter of fact, one of the more interesting piece of work in the paper is related to the characterization of the micro-topography of the landslides and the discussion about the possibility to apply the method to the study of gully erosion. However, gully erosion appear to be as a minor complication compared to the estimation of the landslide volume of a giant landslide. Complication is irrelevant here if the authors focus their paper on the calculation of the total landslide volume.

**<Our Response #2-2>:** We have divided the above section into 2 questions (a to b), and responded the questions accordingly:

a)  "I disagree with the assumption that detailed UAV imagery are better than aerial photographs and/or satellite images to detect and characterized large landslides."

**Response #2-2a:** In Taiwan, heavy precipitation induced by the annual northeast monsoon modifies easily the landslide topography. On the other hand, the study region is situated within a national park and preserves dense forest very well. Both effects conceal detaiedl topography and nearly impossible to study directly from aerial photographs and/or satellite images. The same situation can be found in the two giant landslides (namely, Tsaoling and Jiufengershan) triggered by the Chi-Chi earthquake, where the vegetation colonization concealed almost all the topographic details, especially for the zone of accumulation in just ten years after the landslides occurred. That is why we employed high-resolution and high-precision datasets/methods, the UAV and the airborne LiDAR, to decipher the landslide features of the study area. And that is why we assert the quality levels of the datasets, and illustrated them in Figs. 2 and 4. We have newly revised and clarified the documentation in the manuscript.

b)  "Gully erosion appears to be as a minor complication compared to the estimation of the landslide volume of a giant landslide. Complication is irrelevant here if the authors focus their paper on the calculation of the total landslide volume."

**Response #2-2b:** Yes, the gully incision is a minor factor to estimate the overall landslide volume. The method is used only to assess the landslide morphology and

evolution. We have clarified the documentation in the manuscript.

**<End of Our Response #2-2>**

**#2-3.** The final interpretation is not convincing and rise many question: Why just such two landslides developed along a regional normal fault? What about other places along the fault? There is somethings peculiar in the specific location of the two landslides? (i.e. relative relief higher respect to other places along the fault?) geo-structural setting different respect to other places along the fault and prone to landslides? cluster of strong earthquakes? evidence of high vertical deformation rates? what else?) In the scheme proposed by the authors the fault is the main factor controlling both the onset and the disruption of the landslides, but no analysis support their conclusion. I have also reservation about the idea that normal fault activity has the effect of cancel the landslide signature (third diagram in the final scheme). I think quite the opposite; fault activity sustain relief formation, maintaining the condition for landslide development (see Bucci et al. 2016, ESPL, 41, 711-720; and Densmore et al. 1997, Science 275, 369-72). The authors conclude somethings similar at lines 27-29 page 12, but their statement conflict with the idea illustrated in the scheme. Finally, the authors never explicitly address time scales of the considered landslides and fault, as well as the probable mismatch in timescale of the landsliding and faulting processes.

**<Our Response #2-3>:** We have divided the paragraph into 3 questions (a to c) and responded accordingly:

**a)** "Why just such two landslides developed along a regional normal fault? What about other places along the fault? There is somethings peculiar in the specific location of the two landslides? (i.e. relative relief higher respect to other places along the fault?) geo-structural setting different respect to other places along the fault and prone to landslides? cluster of strong earthquakes? evidence of high vertical deformation rates? what else?)"

**Response #2-3a:** In northern Taiwan, the tectonic activity is in extensional regime, thus dominated by normal faulting in the study area nowadays. The Jinshan fault (JSF), and Shanchiao fault (SCF, also known as the Jinshan Fault with normal faulting mechanism), both of the faulting were being identified longtime ago in Taipei Basin area (Southwest to the study area) and in northeastern offshore

Taiwan (northeast of the study area). And recently these two faults were identified to have linked together as only one normal fault in the Tatun Volcano region around and across the study area by the authors. The result was published in the Central Geological Survey project report written in Chinese, and the paper for international journal is now in preparation. On the other hand, there are many landslides within the study area and in the Tatun Volcano region, as shown in the newly added Fig. 5. Comparing the size, distribution and classification, the two largest landslides (XSL and CSL) were thus chosen as the target for this study. We have revised and improved the paragraph in the manuscript.

**b)** "I have also reservation about the idea that normal fault activity has the effect of cancel the landslide signature (third diagram in the final scheme). I think quite the opposite; fault activity sustain relief formation, maintaining the condition for landslide development. The authors conclude somethings similar at lines 27-29 page 12, but their statement conflict with the idea illustrated in the scheme."

**Response #2-3b:** In northern Taiwan, the tectonic activity of the region is in extensional regime. The Jinshan normal faulting resulted in the formation of Taipei basin by over one thousand meter throw of the fault separation. The normal faulting has been very well documented recently, e.g. Teng et al., (2001); Shyu et al., (2005); C.T. Chen et al., (2007, 2010); Huang et al., (2007); and K.C. Chen et al., (2010). And this normal faulting may also cause the continuous eruption of the Tatun Volcano. The evidence of normal faulting has been recently identified in Taipei basin area and northeastern offshore Taiwan. And two original normal faults are considered to be linked together as a long stretched normal fault that may provide significant earthquake faulting. Finally, the total length of the Jinshan normal fault is more than 130 Km long. We thus interpret that the normal faulting has led to the formation of the slope daylight, as well as the volcano subsidence in the south of the study area. This process may likely lead to the formation of the landslide. Because the normal faulting activated continuously, the sliding mass may be transporting continuously to the Jinshan Delta. The original Fig. 13 (now modified as Fig. 14) demonstrates the general geomorphologic evolution ideally, so as to explain the wear off of the landslide deposits, especially in XSL.

**c)** "The authors never explicitly address time scales of the considered landslides and fault, as well as the probable mismatch in timescale of the landsliding and faulting."

**Response #2-3c:** The normal faulting started from 400 Ka and activated continuously ever since. The age of the Tatun volcano is smaller than 200 Ka. So the relative age of the landslide is most probably after the normal faulting and the formation of the Tatun Volcano, which is later than 200 Ka. On the other hand, the CSL and XSL preserve different degrees of landslide geomorphologic components, showing a combination of multiple landslide events. Furthermore, part of the fault branches is identified on the lower slope within the sliding area, prompting the faulting behavior truncates and enhances the erosion process. In conclusion, based on many aspects, the authors thus propose one model to highlight the possible landslide evolution that will be useful for further testing.

**<End of Our Response #2-3>**

**#2-4.** Finally, I have reservation about the general organization of the paper. The chapter Introduction is a blend (sometime confused) of general issues about landslide identification and characterization. I suggest to restructure the text, developing a sharper motivation with some clearer objectives. Also, quote the pertinent literature addressing the mapping and analysis of large landslides. Pertinent local literature help understanding the state of the art at local scale. The authors are not clear enough at this point. For instance at line 25 page 2 the authors acknowledge that the two landslides were already recognized. So why the authors define the two landslides as "obscure" if they were already recognized? I think additional information should be provided, and a comparison of previous and new results should be done. Similarly, the manuscript lacks of references to international literature addressing mapping and analysis of large landslide in active regions. Pertinent international literature help defining the framework of the study and it should be quoted along the paper (see for instance Bucci et al. 2016, ESPL, 41, 711-720; Scheingross et al. 2013, Geological Society of America Bulletin, 125, 473-489; Bucci et al. 2013, Physics and Chemistry of the Earth, 63, 12–24; Strecker M.R. and Marret R. 1999, Geology, 27, 307-310) The chapter geological background (lines 14-23 page 3) is confused: it is hard to follow and to understand the polygenic history of the faults of the area. The chapter contain information negligible for the aim of the paper. At the same time, the chapter lack of potentially useful information about the age and deformation rate of active structures, seismicity, landslide events. Finally, lines 3-11 page 4 belong to method, not to geological background. The chapters 3 and 4 mix up methods, results and discussion, which is also included in the following chapter: Discussion. This writing setting makes reading hard to follow and to understand. Please change the text of the manuscript including the following chapters: Methods (include here technical issues

regarding UAS imagery, digital topography (1 x 1 m LiDAR DEM), how you define landslides, what do you map using conventional approach (i.e. stereoscopic aerial photo-interpretation), what new using UAS imagery and LiDAR DEM (would be good to see in map the differences), how you estimate the landslide dimension, how you carried out the morphological reconstruction); Results (includes the new data and maps); and then Discussion (what can we learn from the new data and what is the meaning also comparing to other works) and Conclusions (take home messages in short). The chapters Discussion and Conclusion focus on the evolution of the two landslides, stressing the role of tectonics. However, the paper do not contain any new information/analysis/result related to tectonics. The evolution scheme drawn by the authors remain poorly constrained also by the lacks of geological evidences supporting the supposed coincidence of the slip surfaces and the present day topography. I suggest to reconsider in depth (or to drop) the part of the analysis related to the volume calculation of the two landslide, because it simply raises too many questions.

**<Our Response #2-4>:** We have divided the above paragraph into 4 questions (a to d), and responded accordingly:

a) "I suggest to restructure the text, developing a sharper motivation with some clearer objectives. Also, quote the pertinent literature addressing the mapping and analysis of large landslides. Pertinent local literature help understanding the state of the art at local scale. The authors are not clear enough at this point. For instance at line 25 page 2 the authors acknowledge that the two landslides were already recognized. So why the authors define the two landslides as "obscure" if they were already recognized? I think additional information should be provided, and a comparison of previous and new results should be done. Similarly, the manuscript lacks of references to international literature addressing mapping and analysis of large landslide in active regions. Pertinent international literature help defining the framework of the study and it should be quoted along the paper."

**Response #2-4a:** Pertinent literatures are now added into the manuscript. The two landslides were already recognized from 40 m DTM by Prof. C. T. Lee of the National Central University from only personal communication. However, due to the lack of available datasets and without distinct features, the landslides were not analyzed in depth till this study. From climatologic point of view, the annual rainfall is more than 2500 mm in this area, thus a vast portion of the study area is

covered by vegetation. Dense forest thus partially conceals morphological features and has prevented detailed geomorphic studies in the past. On the other hand, the heavy rainfall also enhances the surface processes, e.g., incision and erosion. As a consequence, the erosion effect also obscures the landslide features. We have newly improved the documentation in the manuscript based on the abovementioned points.

b) "The chapter geological background (lines 14-23 page 3) is confused: it is hard to follow and to understand the polygenic history of the faults of the area. The chapter contain information negligible for the aim of the paper. At the same time, the chapter lack of potentially useful information about the age and deformation rate of active structures, seismicity, landslide events."

Response #2-4b: To discuss the landslide evolution, especially for an old landslide, the geologic and regional tectonics must be included. The polygenic history of the study area must be taken into account. In the study area, we consider many factors, including, lithology, normal fault, climate, vegetation, erosion and human agriculture activity etc., in order to access the landslide geomorphologic evolution. Regarding the slip rate of Jinshan normal faulting, it is shown between 8.2-1.8 mm/yr subsiding rate at different sites and in time intervals (e.g., Rau et al., 2006; Huang et al., 2007; Chen et al., 2010). This high slip rate creates the Taipei Basin, and may significantly affect the landslide evolution as well. But unfortunately, these slip rate studies were focused only on the Taipei Basin, and not on the study area. The manuscript is now reinforced and improved to clarify the tectonic factor and the interaction.

c) "This writing setting makes reading hard to follow and to understand. Please change the text of the manuscript including the following chapters: Methods; Results; and then Discussion and Conclusions."

Response #2-4c: We have newly improved the manuscript according to the comment.

d) "The chapters Discussion and Conclusion focus on the evolution of the two landslides, stressing the role of tectonics. However, the paper do not contain any new information/analysis/result related to tectonics. The evolution scheme drawn by the authors remain poorly constrained also by the lacks of geological evidences supporting the supposed coincidence of the slip surfaces and the present day

topography."

Response #2-4d: One geological map (Fig. 5) has been added to demonstrate the geological background of study area, and to better link the relationship between the regional tectonics and landslide geology and evolution. The manuscript has been improved accordingly.

<End of Our Response #2-4>

**End of Comments by Anonymous Referee #2**

---

## Author Response (AR2)

We thank the Editor for the comments that allowed us to clarify and improve the contents of the manuscript. Our answers to the Editor's comments are listed by a point-by-point basis as follows. The green text in the manuscript indicates the revisions that we made in response to the two referees' comments. The blue text in the manuscript indicates the changes that we further made after the latest comments by the Editor.

**Comments by Editor Hayakawa**: The authors have corrected their manuscript following the reviewers' suggestions. There are some more rooms to revise it as listed below.

**NE#1:** #1-2 and #2-1: The authors now showed that both the UAS-derived DSM and ALS-derived DTM are used for the geomorphological analyses. However, in some cases, the descriptions remain unclear. Therefore, I would like to recommend to further revise the manuscript text and figures to clarify which data were actually used for each analysis. For instance, Figures 6 and 7 provide a hillshade image by ALS-DTM, but the authors state that UAS-DSM is also used for the landslide body area. The data source for profiles in Figure 9 must be from ALS-DTM (I guess), but it is not explicitly shown. By clarifying the more detailed morphological characteristics of the landslide (even if they are significantly modified), it will become possible to argue the issues on the largeness of the landslide size (#2-3a), which can be due to multiple occurrences of sliding.

**Answer:** Thank you for the comments. We have deleted original Fig. 8 and added a comprehensive new figure, now Fig. 7, to demonstrate the datasets used for this study. The text in the manuscript is also revised accordingly. The data source for profiles in now Fig. 8 is also indicated as did for other figures (Figs. 10, 11, 12) that were not clearly indicated in previous version.

**NE#2:** Related to the issue above, Figure 6 only shows the hillshade data by ALS-derived DTM. Is it possible to overlay the hillshade by UAS-derived DSM onto this image? Or, as noted below, adding some magnified views of the key morphological features (somewhat like the case in Figure 4) to this or next figure is another option.

**Answer:** Similar to the previous comment, we have revised relevant figures in the text. Actually, we used the UAS imageries to create true 3D models, which are the most useful product from UAS system. Regarding Fig. 6 (now Fig. 5), we did not modify the figure. Instead, in order to better illustrate the morphologic features, we newly created Fig. 7 for the suggested purpose. This also responds to the comment of

NE#3.

**NE#3:** Figure 8 is only mentioned once in the text. This anaglyph may be somewhat useful but not fully because the geomorphological details must have been assessed using the UAS-derived DSM, not only the ALS-DTM. This figure can therefore be omitted. Instead, I would recommend adding more detailed views of key features (main scarp, ridge, etc.) in Figure 7 with their magnified views by UAS-DSM.
**Answer:** The original Fig. 8 is now deleted and we have added a new figure (now Fig. 7) to conform the recommendation.

**NE#4:** The terms DTM and DEM are mixed. In general, DEM is often an inclusive term, while DTM is more specifically used to indicate the land surface (terrain) data after filtering. I would recommend defining the terms of DSM and DTM when they appears firstly in the manuscript, and replace DEM with DTM thereafter.
**Answer:** We have now defined the terms in P. 5, l. 31 to P. 6, l. 4. in this manuscript as suggested. (Note: The Taiwan government and the geomatics community define the term DTM as a general term, whereas the DEM was defined as the geomorphologic elevation after removing the buildings, trees and vehicles, etc. DSM defines the first return of the LiDAR pulse of terrain data, including buildings and tree canopy.)

**NE#5:** #2-1e: I could not find the detailed descriptions regarding the relative age of the landslide in the main text.
**Answer:** We have revised the text regarding the time scale issue for landsliding and faulting. The text is distributed within mainly the Discussion sections 5.1 and 5.3. To be more specific and focused, we added a new paragraph in P. 13, l. 7-14 to clarify the comment as follows: "To conclude the landslide generation, the normal faulting in the region started from 400 Ka and is activated continuously ever since. The faulting was identified in the Taipei basin area and northeastern offshore Taiwan, with the fault line situated on both sides of the study area (Figs. 4 and 5). And the fault line was recently identified and linked together as only one normal fault in Tatun Volcano region. In conclusion, for the relative age of the landslide, we interpret that the landslide has been triggered since right after normal faulting started and the formation of Tatun Volcano, which is far later than 200 Ka. Regarding to the different generation of landslide, the geomorphologic components also show different degrees of preservation within the two observed landslides. Furthermore, the CSL is interpreted to have occurred from a combination of multiple landslide events."

**NE#6:** #2-2a: I could not find the relevant documentation in the manuscript.

**Answer:** The revised text is located in P. 4, l. 21 to P. 5, l. 4. Here we would like to emphasize our previous response to this question. In Taiwan, heavy precipitation induced by the annual northeast monsoon modifies easily the landslide topography. On the other hand, the study region is situated within a national park and preserves dense forest very well. Both effects conceal detailed topography and nearly impossible to study directly from aerial photographs and/or satellite images. The same situation can be found in the two giant landslides (namely, Tsaoling and Jiufengershan) triggered by the Chi-Chi earthquake, where the vegetation colonization concealed almost all the topographic details, especially for the zone of accumulation in just ten years after the landslides occurred. That is why we employed high-resolution and high-precision datasets/methods, the UAV and the airborne LiDAR, to decipher the landslide features of the study area. And that is why we assert the quality levels of the datasets, and illustrated them in Figs. 2 and 4. We have newly revised and clarified the documentation in the manuscript.

**NE#7:** #2-2b: I could not find the relevant documentation regarding the significance of gullies in the manuscript.

**Answer:** The revised and newly added text is in P.13, l. 29 to P. 14, l. 2. We would like to stress that the gully incision is a minor factor to estimate the overall landslide volume. The method is used only to assess the landslide morphology and evolution. We have clarified the documentation in the manuscript.

**NE#8:** #2-3a: The authors revised the descriptions, but still it is unclear why the two landslides are so large. In page 12 lines 16-17, the authors seem to argue that the multiple occurrences of sliding resulted in the extremely large landslide, but in page 13 lines 20-21, the total volume is compared with the previously-reported landslides. As noted before, detailed morphological characteristics within the landslide body using UAS-DSM should be provided to further argue this issue.

**Answer:** Please refer to the revised and some newly added text in P. 7, l. 19-26. We have also created a brand new figure shown as Fig. 7. Regarding why such large landslides occurred, as we have emphasized in the Discussion section, it is likely to be resulted from the active normal faulting and rapid erosion process in the region. Regarding the UAS DSM, the dataset is useful for detailed ground identification and validation, however it is easily concealed by dense vegetation. That is why it is necessary to integrate several methods to decipher the landslide evolution as shown in the manuscript.

**NE#9:** (#2-3a cont.) Moreover, the way of reproduction of the geological map (new Fig. 5) is unclear. Did the authors performed field validation, or just re-draw the existing geological map on the ALS-DTM image? (P7, L18-20) Please clarify.

**Answer:** We have clarified the comment by revising and adding a paragraph in P. 7, l. 19-26 as follows: "A detailed regional geologic map was reproduced by investigation from the high-resolution DEMs prior to the landslide study, as illustrated in Fig. 4. The geological structures, e.g. lineament, fault, fold and especially the landslides, are interpreted directly from LiDAR DTM and DSM, and UAS DSM and 3D model, then validated in the field. The color patches represent strata boundaries initially from existing geologic maps, and they were then modified with LiDAR and UAS data (Yeh et al., 2014 and 2017), and again validated in the field if possible. The components contain faults, lineaments and landslides shown by scarps (Fig. 4). For the first appearance, there many landslides occurred in the study area and in the Tatun Volcano region. Comparing the size, distribution and classification, data accessibility especially for UAS flight mission, the two largest landslides (XSL and CSL) were thus chosen as the target for this study. "

**NE#10:** #2-3b: I could not realize what "slope daylight" indicates. Also, the last part of this response is unclear. The newly added description in page 15 lines 18-21 is uncertain: What does "erosion by alluvial processes at the slope toe"? Does this mean that the base-level lowering by fluvial erosion or fault displacement, or both? Please clarify.

**Answer:** The "slope daylight" is a civil engineering term, frequently used for slope stability study and we have added a relevant reference (Yeh, et al., 2017) in the sentence. Regarding the slope toe erosion, graben subsidence is considered a major contributor for the observed phenomena. We have modified the text in P. 16, l. 14-18 to clarify what we meant as follows: "New tectonic activities because of normal faulting will likely result in the reactivation of the existing landslides posing life-threatening situations, particularly if the slope toe is being eroded by alluvial processes transporting sediments downstream and by graben subsidence daylighting the dip slope and reactivating landslide."

**NE#11:** #2-3c: I could not find the relevant documentation in the manuscript.

**Answer:** Overall, this comment is similar to the comment NE#5. Please refer to the answer for NE#5.

**NE#12:**#2-4a: The citation to this finding by Lee seems missing in the manuscript. Please provide an appropriate citation (around P2, L27-28?).

**Answer:** Because the finding was not officially documented in the literature, we have revised and added the following sentence in the text to clarify the citation question: "Two large-scale landslides have already been suggested by C. T. Lee of National Central University (personal communication) from LANDSAT images and 40m grid digital terrain model of the region but without existing documentation in the literature."

**NE#13:** #2-4a: I could not find the relevant documentation in the manuscript.

**Answer:** We have changed the subtitles of the manuscript and reorganized the text which now differs from the first submitted version. The changed subtitles are as follows: Introduction, Geological background, Methods of data acquisition, Results of morphological analysis, Discussion, and Conclusions. We also stress that pertinent literatures are now added into the manuscript. Due to the lack of available datasets and without distinct features, the studied landslides were not analyzed in depth till this study. From climatologic point of view, the annual rainfall is more than 2500 mm in this area, thus a vast portion of the study area is covered by vegetation. Dense forest thus partially conceals morphological features and has prevented detailed geomorphic studies in the past. That's why this study is crucial to reveal the true characteristics of the unusually large landslides. On the other hand, the heavy rainfall also enhances the surface processes, e.g., incision and erosion. As a consequence, the erosion effect also obscures the landslide features. We have improved the documentation in the manuscript based on the abovementioned points.

**NE#14:**#2-4c: Please follow the reviewer's suggestion regarding the chapters.

**Answer:** We have changed the subtitles of the manuscript and revised the text which now differs from the first submitted version. The changed subtitles are as follows: Introduction, Geological background, Methods of data acquisition, Results of morphological analysis, Discussion, and Conclusions.

**NE#15:** Please follow the styles of NHESS, in particular the references (e.g., P14, L8-9).

**Answer:** We have revised the mentioned references and also checked other references to fit the journal's text styles.

**NE#16:** North arrows are missing in Figures 3b-c, 4, 7, 9 (inset), 11, and 12 (inset). That in Figure 6 has an error. The north direction of Figure 8 may also be incorrect.

**Answer:** We have revised north arrows in the figures as suggested. The errors in the mentioned figures are updated. The old Fig. 8 is now deleted as suggested.

**NE#17:** Scales are missing in Figures 3b-c, 4f,h,i, 7, and 9 (inset, unknown numbers are shown in the frame).
**Answer:** We have added the scales in the figures and revised the numbers in the frame.

**NE#18:** Figure 1: Show the summit location and extent area of Tatun Volcano. Please show the extent areas of Figures 3a, 4a-c, 5, and 6.
**Answer:** We have labeled two main summits, Mts. Chising and Tatun, in the figure and the Tatun Volcano Group is indicated by the warm colors, which we think is enough for the purpose of the paper. We also have shown the extent areas of figures where appropriate and necessary as suggested by the comment.

**NE#19:** Figure 2: I do not see this figure is necessary for this paper. The hillshade image (inset) is already shown in Figure 1, while the IKONOS image does not provide clear appearance of the landslides studies. I would recommend simply removing this figure.
**Answer:** We have deleted the figure as suggested.

**NE#20:** Figure 3: Please show the location and direction of panels b and c in a.
**Answer:** We have revised and shown the features in the figure as suggested.

**NE#21:** Figure 5: Does "study area" mean the extent of Figure 6? If so, please show it as Figure 6.
**Answer:** The study area means Figure 6. We have revised accordingly.

Thank you very much for your time and comments for improving the manuscript.

---

## Author Response (AR3)

We thank the Editor for the comments that allowed us to clarify and improve the contents of the manuscript.

**Comments by Editor Hayakawa**: Figure 7: Please re-order the panels c-g in the left-to-right, top-to-bottom direction (following the "Gutenberg Diagram").

**Answer:** We have re-ordered the Figure 7 as suggested.

Thank you very much for your time and comments for improving the manuscript.